# Corticosterone pattern-dependent glucocorticoid receptor binding and transcriptional regulation within the liver

**Benjamin P. Flynn** [1]*, **Matthew T. Birnie** [1], **Yvonne M. Kershaw** [1], **Audrys G. Pauza** [1], **Sohyoung Kim** [2], **Songjoon Baek** [2], **Mark F. Rogers** [3], **Alex R. Paterson** [1], **Diana A. Stavreva** [2], **David Murphy** [1], **Gordon L. Hager** [2], **Stafford L. Lightman** [1], **Becky L. Conway-Campbell** [1]

1 Henry Wellcome Laboratories for Integrative Neuroscience and Endocrinology, University of Bristol, Bristol, United Kingdom, 2 Laboratory of Receptor Biology and Gene Expression, National Cancer Institute, National Institute of Health, Bethesda, Maryland, United States of America, 3 Intelligent Systems Laboratory, University of Bristol, Bristol, United Kingdom

* ben.flynn@bristol.ac.uk

**Data Availability Statement:** Data is available at GEO with accession number GSE171647 (https://www.ncbi.nlm.nih.gov/geo/query/acc.cgi?acc=GSE171647).

## Abstract

Ultradian glucocorticoid rhythms are highly conserved across mammalian species, however, their functional significance is not yet fully understood. Here we demonstrate that pulsatile corticosterone replacement in adrenalectomised rats induces a dynamic pattern of glucocorticoid receptor (GR) binding at ~3,000 genomic sites in liver at the pulse peak, subsequently not found during the pulse nadir. In contrast, constant corticosterone replacement induced prolonged binding at the majority of these sites. Additionally, each pattern further induced markedly different transcriptional responses. During pulsatile treatment, intragenic occupancy by active RNA polymerase II exhibited pulsatile dynamics with transient changes in enrichment, either decreased or increased depending on the gene, which mostly returned to baseline during the inter-pulse interval. In contrast, constant corticosterone exposure induced prolonged effects on RNA polymerase II occupancy at the majority of gene targets, thus acting as a sustained regulatory signal for both transactivation and repression of glucocorticoid target genes. The nett effect of these differences were consequently seen in the liver transcriptome as RNA-seq analysis indicated that despite the same overall amount of corticosterone infused, twice the number of transcripts were regulated by constant corticosterone infusion, when compared to pulsatile. Target genes that were found to be differentially regulated in a pattern-dependent manner were enriched in functional pathways including carbohydrate, cholesterol, glucose and fat metabolism as well as inflammation, suggesting a functional role for dysregulated glucocorticoid rhythms in the development of metabolic dysfunction.

## Author summary

Adrenal glucocorticoid hormones are released in a characteristic ultradian rhythm that becomes dysregulated during chronic stress, disease, or synthetic corticosteroid treatment.

**Funding:** This research was funded in part, by the Wellcome Trust 089647/Z/09/Z awarded to B.L.C-C & S.L.L. and the United kingdom research Council grant MR/R010919/1 supporting B.L.C-C & B.P.F.. Engineering and Physical Sciences Research Council (EP/K008250/1) grant awarded to Dr. Colin Campbell and D.M. and the Needham Cooper Trust PhD scholarship awarded to B.P.F.. For the purpose of Open Access, the author has applied a CC BY public copyright licence to any Author Accepted Manuscript version arising from this submission. The funders had no role in study design, data collection and analysis, decision to publish or preparation of the manuscript.

**Competing interests:** The authors have declared that no competing interests exist.

Metabolic dysfunction is a comorbidity associated with all these conditions, but the role that altered glucocorticoid dynamics play is unknown. As the liver is a major site of glucocorticoid action on metabolic homeostasis regulated by the glucocorticoid receptor, we have assessed how different patterns of hormone replacement in adrenalectomised rats differentially regulate gene pathways involved in type II diabetes, cirrhosis, and fatty liver development, via altering the pattern of glucocorticoid receptor binding to regulatory sites. We believe our findings have important implications for therapies that can reproduce the endogenous glucocorticoid rhythm and thus minimize adverse metabolic side-effects in patients.

## Introduction

In mammals, glucocorticoid (GC) release from the adrenal glands is under the regulation of the hypothalamic pituitary adrenal (HPA) axis [1]. Entrained to the photoperiod, and regulated by the hypothalamic suprachiasmatic nucleus [2], maximal GC release occurs prior to awakening and gradually decreases throughout the active period before returning to basal levels at the onset of the inactive period, in a classical 24 hr circadian profile. The vast majority of GC rhythm research has focussed on this circadian profile using 6–12 hr data collection intervals, however, automated blood microsampling at 10 min intervals has led to the detection of a pronounced underlying rhythm of rising and falling phases with a periodicity of 56–60 min for corticosterone (CORT) in rat [3] and 95–180 min for cortisol in man [4,5]. The origin of the ultradian rhythm within the circadian rhythm has been shown both mathematically and experimentally to emerge as a consequence of a sub-hypothalamic pulse generator, where intrinsic delays in pituitary adrenal communication exist in both the positive feed-forward and negative feedback arms of the loop [6]. As GC oscillations are highly conserved across mammalian species [7], and bioavailable 'free' GC pulses can be detected by microdialysis in the extracellular fluid of a variety of tissues [8,9] in experimental rodents and in man [10,11], we propose that this dynamic hormone signal may be crucial for optimal tissue specific responses to GC secretion throughout the body.

Adrenal GCs regulate multiple physiological processes via regulation of glucocorticoid receptors present in almost all tissues of the body, thus affecting immunological and stress responses, cognitive processes, cardiovascular regulation, and a range of metabolic processes including glucose, lipid, fatty acid and triglyceride homeostasis. GCs promote gluconeogenesis in the liver, whereas they decrease glucose uptake and utilisation in skeletal muscle and white adipose tissue by antagonising insulin response. Excess GC exposure causes hyperglycaemia and insulin resistance. GCs also regulate glycogen metabolism [12]. Associations between elevated circulating GCs and metabolic disease are well known. For example, Cushing's disease, which is characterised by hypercortisolism, is associated with a myriad of metabolic co-morbidities including increased visceral fat deposits, obesity and dyslipidaemia; furthermore, type II diabetes occurs in at least a third of Cushing's patients [13]. Synthetic GCs (sGCs), which are often more potent and have longer plasma half-lives than their endogenous counterparts [14,15], can cause a similar pattern of adverse metabolic side effect. Patients prescribed GC treatment have a higher incidence of metabolic disease, including diabetes, compared to patients on other medications [16–19]. There has been considerable investigation of the mechanisms underlying these side effects over the past decade and whilst dose is certainly a well-recognised risk factor, efforts to reduce doses towards physiological levels is often insufficient to prevent the development of metabolic disturbance over a prolonged period [20]. GCs regulate

gene transcription in target cells via a ligand activated transcription factor, the glucocorticoid receptor (GR). Activated GR is directed to specific sites across the genome, either by direct binding to short consensus DNA sequences termed glucocorticoid response elements (GREs) or via tethering by other transcription factors. GR then recruits co-factors such as the histone acetyl transferase enzymes CBP/P300 and the ATP-dependent Swi/Snf chromatin remodelling complex to reposition nucleosomes and allow increased accessibility for RNA polymerase II (RNAPol2) and related transcriptional machinery to the transcriptional start site (TSS) of target genes [21,22]. Conversely, decreased access of RNAPol2 to the TSS of target genes results in transcriptional down-regulation. In this way, GCs can induce either up-regulation or down-regulation of a vast number of target genes in a highly gene-specific, cell-specific and context-specific manner.

Studies using high-dose pulsed CORT treatment in adrenalectomised rats [23] have demonstrated that GR binds transiently to GREs regulating the transcription of the 'hypersensitive' *Period 1* gene [24] in multiple cells and tissues throughout the body and brain [14,25,26]. In rat liver, GR binding at *Per1* regulatory GREs reached maximal levels at the peak of an hourly induced CORT pulse before fully dissociating within the 40 min period of CORT clearance from the circulation [14]. Despite the supraphysiological CORT dose, synchronised recruitment and loss from the DNA template was repeated over the series of pulses and was found to induce similarly synchronised, if slightly delayed, enrichment of RNAPol2 at the *Per1* TSS and hnRNA production within liver [14] as well as brain tissue [25] which was consistent with observations in a variety of cell lines [14,25–28]. More recently, assessment of genome-wide GR binding and RNAPol2 enrichment at the TSS in a murine mammary cell line (3617) revealed recruitment and loss synchronised to ultradian corticosterone replacement patterns at a vast number of targets, and these dynamics were ablated by constant CORT treatment [28]. What has remained unknown, until now, is how these genome-wide dynamics are affected during ultradian versus constant CORT exposure in a live animal. Furthermore, as it is now well accepted that cell-specific chromatin architecture will pre-determine a different GC-target transcriptome in liver compared to 3617 cells, it is extremely important to assess how transcriptional regulation of physiologically relevant GC-target genes associated with metabolic function are affected by disrupted GC exposure and how this may relate to metabolic function / dysfunction in the whole animal.

Here, we demonstrate that genome-wide GR binding dynamics in liver are tightly regulated by a pulsatile CORT infusion (PULS). In contrast, a dose-matched constant CORT infusion (CONS) results in widespread disruption of ultradian GR dynamics, with subsequent impact upon RNAPol2 recruitment to liver-specific GC regulated genes and significant pattern dependent effects on the GC regulated liver transcriptome. We have identified key metabolic targets that are sensitive to altered CORT exposure, thus highlighting metabolic pathways that are acutely susceptible to GC dysregulation, with implications for the development of the clinically important metabolic syndrome in patients.

## Results

### Validation of pulsatile and constant corticosterone infusion in adrenalectomised rats

To model a physiologically realistic ultradian CORT replacement pattern in adrenalectomised rats, we based our PULS on the endogenous CORT profile during the peak secretory phase, as determined by frequent blood sampling of adrenally intact male rats at 10 min intervals over a 28 hr period (Fig 1A). As CORT pulses are not synchronised between rats (Figs 1B and S1A–S1E), the underlying ultradian rhythm is only evident within individual CORT profiles (Fig

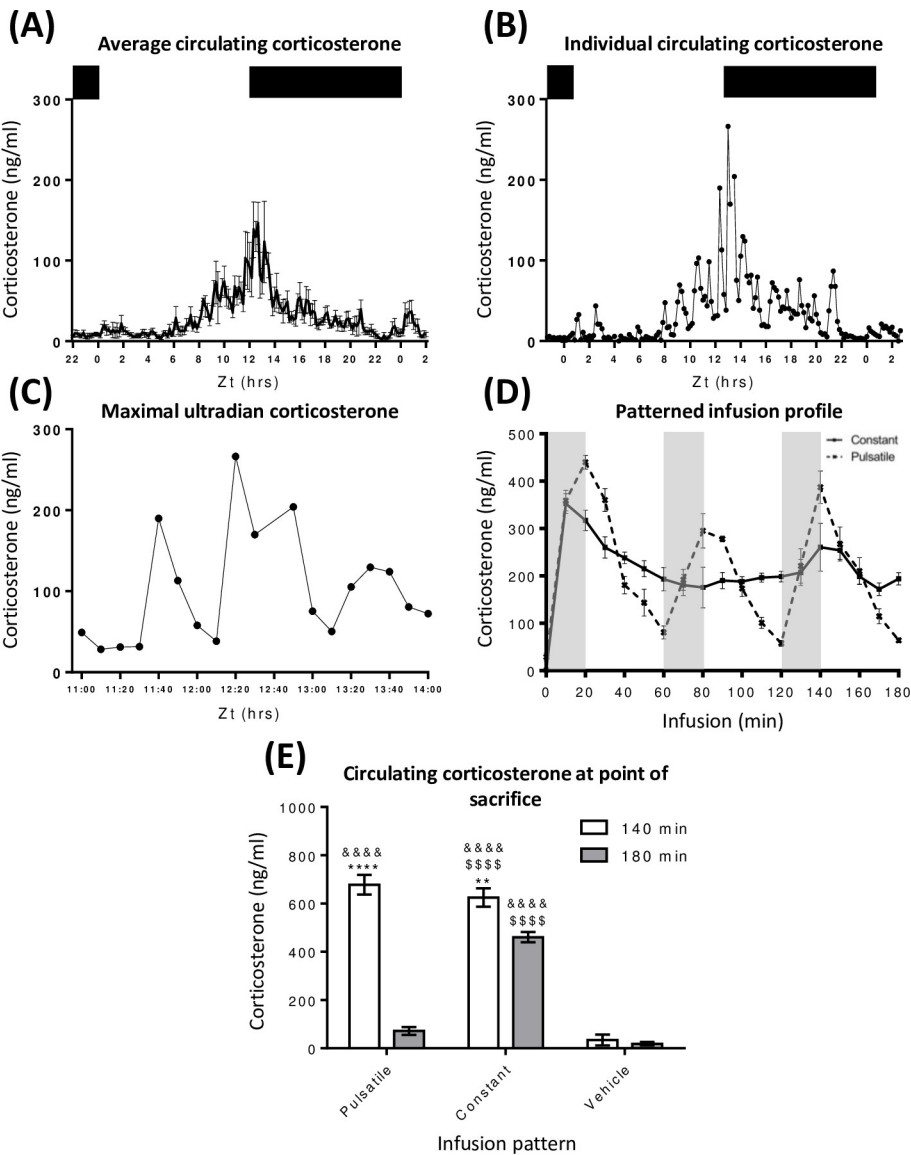

**Fig 1. Validation of both the dose and infused patterns of CORT within the circulation of the rat.** (A) Blood samples were taken every 10 min over a 28 hr period. CORT levels gradually increased from basal (Zt 0–6) to maximal levels at Zt 12:40. Levels decreased from this point returning to baseline by Zt 22. Lights were 12:12 (lights off at Zt 12), N = 6 and data is expressed as mean ± s.e.m.. (B) Circulating CORT levels from a single rat indicate approx. hourly oscillations in CORT in a classical ultradian profile which are (C) most discernible during maximal circadian secretions, as indicated between Zt 11–14. N = 1. (D) 1 ml of CORT (3.84 μM) was delivered intravenously over 3 hrs in a PULS manner (dashed line) via 20 min (grey bar), hourly CORT infusions at a rate of 1 ml/ hr, whilst the CONS (solid line) rate of 0.33 ml was maintained throughout the experimental period. Peaks in circulating CORT were observed at 20 min, 80 min and 120 min (353 ng/ ml ± 42) post each 20 min infusion and was cleared from the circulation within each 40 min pump cessation at 60 min, 120 min and 180 min (68 ng/ ml ± 20). CONS-induced sustained circulating levels (222 ng/ ml ± 12) between 10–180 min of infusion. Area under the curve analysis confirmed no difference in dose delivered per hr. N = 4 and mean ± s.e.m.. (E) Total circulating CORT was measured from trunk blood samples taken at the point of euthanasia (140 min and 180 min) after either PULS, CONS or VEH. Two-way ANOVA indicated a significant effect of time, infusion pattern and interaction (p < 0.0001) in response to infusions. Post-hoc tests indicated significantly raised CORT at PEAK_140 min compared to both NADIR_180 and VEH (p < 0.0001) time points. In response to CONS, CORT was raised at both 140 min and 180 min compared to NADIR_180 and VEH (p < 0.0001), but there was a significant reduction in CORT from 140 min at 180 min (p < 0.01). Lights were 12:12 (Zt 12 lights off as indicated by the top black bar) and represents total blood serum CORT levels. N = 7 and mean ± s.e.m. and significant comparisons between CORT infused and VEH infused matched time points indicated by &&&& p<0.0001, between CORT infused time points by ** p<0.01 and **** p < 0.0001 and between infusion patterns by $ $ $ $ p < 0.0001.

1B) and most marked during maximal circadian CORT secretion (Fig 1C). To recapitulate these endogenous ultradian CORT rhythms, 20 min CORT infusions were delivered hourly. Blood sampling at 10 min intervals showed CORT levels rising for 20 min, before falling rapidly as CORT was cleared from the circulation. This pattern was robustly repeated over the 3 hr experimental time course, averaging 374 ± 24 ng/ ml at pulse peak (20 min into each hour) and 64 ± 6 ng/ ml at pulse nadir (every 60 min) (Fig 1D).

CONS was dose matched to PULS and confirmed by area under the curve analysis. After the initial transient increase between 0–20 min in circulating CORT (maximal 353 ng/ ml ±42), levels remained relatively stable between 60–180 min (201 ± 51 ng/ ml) (Fig 1D). Liver samples were taken at exact times corresponding to the third pulse peak (140 min) and nadir (180 min) of the PULS. CONS and vehicle control infusion (VEH) samples were taken at the same time points. Circulating CORT levels were confirmed for all rats taken for liver chromatin immunoprecipitation assay (ChIP) processing (Fig 1E).

## Genome-wide GR binding is glucocorticoid pattern dependent

Liver samples were assessed for GR binding across the genome using ChIP assay followed by next generation sequencing (ChIP-Seq). In total, 3,098 sites of significantly increased GR intensity (relative to VEH) were identified across all groups, indicating sites of binding. Within these sites, striking CORT pattern-dependent and time-dependent changes were observed (Fig 2A–2F). Differences in tag density were evident at a subset of sites when comparing 140 min pulsatile corticosterone infusion (PEAK_140), 140 min constant corticosterone infusion (CONS_140) and 180 min constant corticosterone infusion (CONS_180), but all three groups exhibited an overall increase in tag density enrichment relative to both 180 min pulsatile corticosterone infusion (NADIR_180) and VEH.

Hierarchical clustering (Fig 2G) revealed the global extent of transient GR binding in liver during PULS. Of the total 3,098 identified binding sites, 2,658 were enriched (relative to VEH) at the PEAK_140 time point, whereas none were significantly detected at NADIR_180 (Fig 2H), indicating robust and transient GR binding synchronised to the CORT pulse peak. At CONS_140, approximately half the number (1,273 sites) were enriched (relative to VEH) compared to the matched PULS time point, however, in stark contrast to the complete lack of enriched sites at NADIR_180 to VEH, 1,953 binding sites were detected at CONS_180 (Fig 2G and 2H). Despite the increase in number of enriched sites detected at 180 min compared to 140 min of CONS, only 79 of these were found to be significantly differentially enriched in a direct comparison between time points (Figs 2J and S3C), indicating sustained enrichment over time at majority of sites during CONS.

As shown in the Venn diagram (Fig 2I), the largest proportion of GR binding sites (33.4%), were commonly enriched between PEAK_140, CONS_140 and CONS_180 time points. Almost a third of total sites were uniquely enriched at PEAK_140 (31.0%), whilst just a small proportion were unique to CONS (12.6%), and an even smaller proportion were exclusively associated with either CONS_140 (3.0%) or CONS_180 (7.6%).

Interestingly, a very small number of sites were found to have reduced enrichment of GR (relative to VEH), 50 at PEAK_140 (1.6% of all enriched sites), 10 at CONS_140 (0.3%) and 9 at CONS_180 (0.3%). Tag density distribution plots support these as regions of enrichment indicative of real binding events (S2F–S2J Fig), although a mechanism explaining how GR binding can be greater in adrenalectomised rather than in CORT treated rat liver is currently not well understood.

Together these data show how ultradian GC-dependent GR binding dynamics alter during constant CORT exposure.

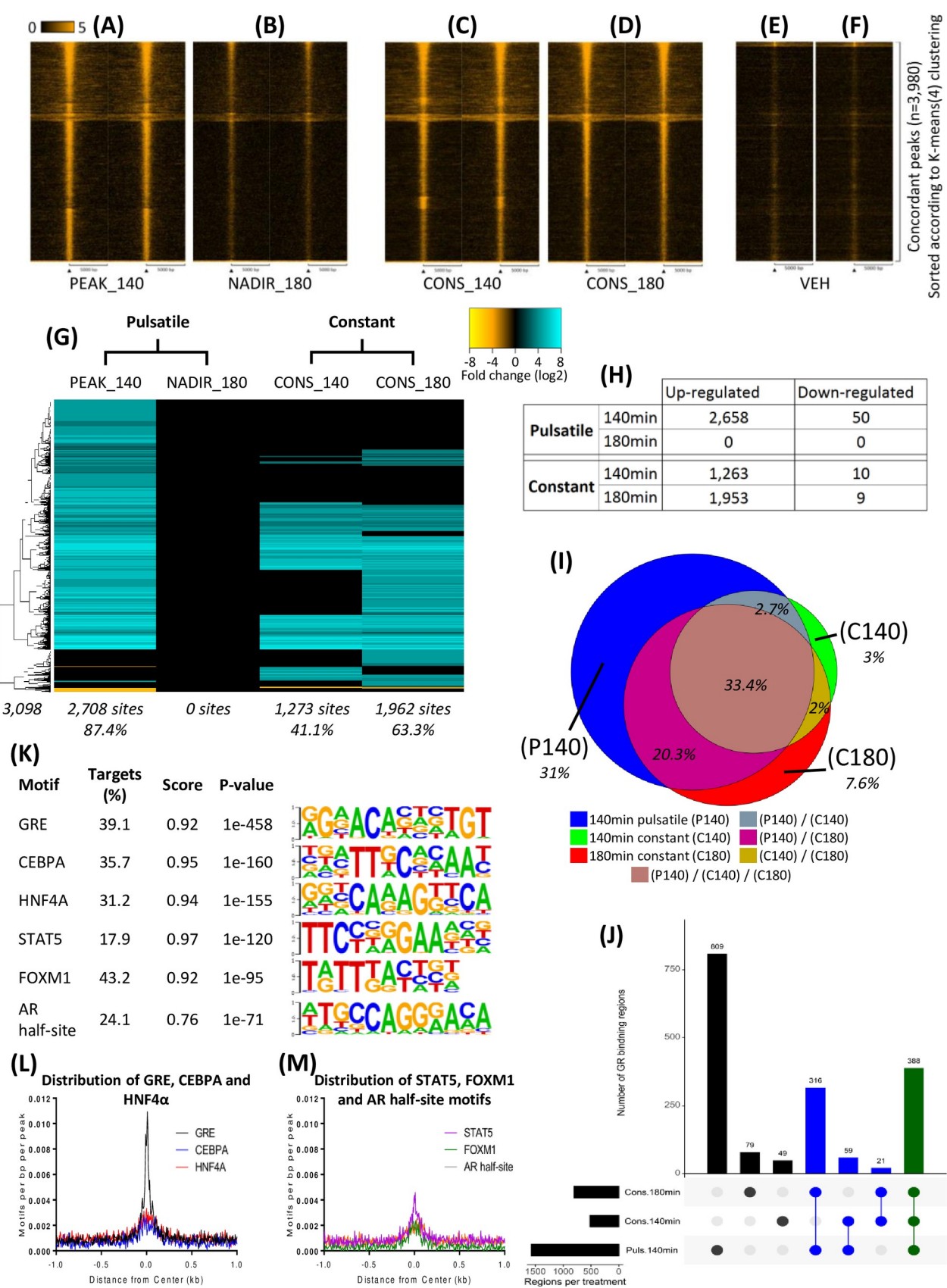

**Fig 2. Patterned CORT-regulated GR binding at sites across the genome.** GR tag intensity and distribution from the centre of 3,980 enrichment regions in response to (A) PEAK_140, (B) NADIR_180, (C) CONS_140, (D) CONS_180 as well as (E) 140 min and (F) 180 min VEH was visualised by heatmap. Tag count intensity was normalised to 1 million reads per 1 kb, over a 10 kb region, segmented into 200 bins and is indicated by the heatmap according to the scale bar. (G) Heatmap indicates GR enrichment fold change within identified regions in the liver in response to either PULS or CONS at 140 min and 180 min compared to VEH. Heatmap indicates merged replicate tags (N = 2), data was hierarchically clustered according to fold change (log2) and colour intensity indicates degree of fold change as indicated by the scale bar between -8 to 8 (top right). Total number of genes differentially regulated is indicated bottom left, number of genes in response to infused CORT time point under each lane as well as the percentage of total genes. (H) Table describing the number of significant CORT-regulated enrichments compared to VEH. (I) Venn diagram representing the number of GR binding sites induced by single or multiple CORT infused time points compared to VEH. (J) Graph represents the number of sites induced compared to NADIR_180 by a single or combination of PEAK_140 and CONS time points. Y-axis of the vertical bar graph indicates number of sites whilst the dot directly below a bar represent if a site was enriched by a single (black), two (blue) or three (green) CORT infused time points. Horizontal bar graph indicates the number of sites enriched by each CORT infused time point. (K) Table of the top 6 significantly over-represented sequences within CORT-regulated GR sites to VEH control. Score indicates degree of similarity between *de novo* and known motif sequences and motif base logo indicates specific probability (s.p.) of base occurrence. (L-M) Of the most significant motifs, histograms report the intensity and distribution of the (L) GRE (black), CEBPA (Blue), HNF4α (red), (M) STAT5 (purple), FOXM1 (green) and AR half-site (grey) motifs 1 kb in either direction from the centre of identified GR enriched regions.

## Lack of evidence for a co-factor that determines differential GR binding dynamics

To identify determinants of liver-specific GR binding, as well as pattern-dependent co-factors, all significantly enriched sites from the GR ChIP-Seq datasets were analysed for over-represented *de novo* motif sequences. Of the top 6, the most confident match (p = e-458) was to the GRE motif (Fig 2K), followed by CEBPA, HNF4a, STAT5, FOXM1 and AR half-site consensus sequences. GRE distribution was most distinctly concentrated at the centre of the identified GR binding regions compared to other motifs (Fig 2L and 2M), further supporting identified enriched regions as direct GR binding sites as well as indicating specificity at the estimated location of GR to DNA interaction. Co-occurrence of HNF4α or STAT5 motifs with GREs was detected in over 50% of GRE-containing regions, while co-occurrence of CEBPA with GREs was detected in over 35% of GRE-containing regions (S4A Fig). As CEBPA has been demonstrated to maintain chromatin accessibility for GR binding in liver [29], together with other transcription factors including HNF4α, our findings most likely reflect these cell type-specific determinants of GR binding in liver rather than their involvement in determining differential responses to patterned GC exposure. The over-representation of STAT5 motifs within GRE-containing regions also likely reflects the known interactions of these two transcription factors in regulating a large number of functional groups of genes in liver [30]. Further interrogation of sequences underlying pattern-dependent GR binding sites was performed to determine whether specific motifs were enriched at sites exclusively induced by either PULS or CONS. Motifs underlying GR binding sites, however, were evenly represented across infusion pattern and times, indicating that differential GR binding patterns are highly unlikely to be determined by associated trans-acting factors (S4B and S4C Fig).

## RNA polymerase II enrichment in target genes is regulated in a pattern-dependent manner

To investigate if the observed changes in CORT pattern-dependent GR binding modulates transcriptional activity, liver chromatin samples were immunoprecipitated using an antibody specific for a phosphorylated serine 2 residue on the carboxy terminal domain of RNAPol2 (pSer2 Pol2). As the pSer2 Pol2 is associated with the actively transcribing complex [31,32], intragenic occupancy data serves as a proxy for active transcription at the level of the chromatin.

pSer2 Pol2 occupancy exhibited dynamic regulation in a CORT time-dependent and pattern-dependent manner that was reflective of the GR binding patterns observed. PEAK_140

induced transcriptional changes in the largest proportion of total regulated genes (553 genes) whilst intragenic pSer2 Pol2 occupancy mostly returned to basal levels at NADIR_180 (for 449 genes) with the exception of 104 genes (Fig 3A, 3D and 3E). Interestingly, for both PEAK_140 and NADIR_180, there was decreased pSer2 Pol2 occupancy compared to VEH in approx. 3 out of 4 cases (Fig 3A and 3D) indicating an overall inhibitory transcriptional effect by PULS, although this was far more pronounced at PEAK_140 due to the transient nature of the vast majority of the changes. During CONS, differential pSer2 Pol2 enrichment relative to VEH was detected at both CONS_140 (254 genes) and CONS_180 (278 genes) (Fig 3A). More genes were up- rather than down-regulated at 140 min (196 and 58 genes respectively), whilst there was a more similar distribution at 180 min (138 and 140 genes respectively) (Fig 3D).

Consistent with observed GR binding dynamics, the most robust effect on transcriptional regulation was detected at PEAK_140, in a highly transient profile with a return to basal levels by NADIR_180 in most cases. In contrast, more moderate changes were detected with CONS, with lower fold change and fewer genes affected and sustained across both time points (Figs 3A, S3D, S3E and S3F).

## Comparative mRNA expression analysis

To assess the impact of the observed highly dynamic patterns of pSer2 Pol2 occupancy to the liver transcriptome, we performed RNA-Seq expression analysis on liver samples collected at 180 min PULS and CONS. The analysis revealed differential mRNA expression (relative to VEH) for 282 of the genes identified as differentially occupied by pSer2 Pol2 (39.6% of ChIP targets) (Figs 3B, 3D and S5). Whilst the direction of change in pSer2 Pol2 occupancy and RNA expression was found to be largely similar across data sets (Figs 3B, 3D and S5), there was an interesting exception; the pronounced transient decrease in pSer2 Pol2 occupancy observed at PEAK_140 did not result in significantly reduced mRNA expression levels at 180 min, for the majority of cases. Transient reduction was observed in 406 of 453 pSer2 Pol2 differentially occupied genes at PEAK_140 and at 180 min, whilst just 41 of the 144 genes expressions were reduced in the RNA. Interestingly, the transient increased pSer2 Pol2 occupancy of 147 genes at PEAK_140 was followed by increased mRNA expression for 103 genes at 180 min (Figs 3C and S5). Whilst in stark contrast, CONS decreases in pSer2 Pol2 occupancy between 140–180 min were faithfully accompanied by a significant down-regulation of mRNA at 180mins (146 genes), as was an approx. equal number of up-regulated genes (121 genes) compared to VEH (Figs 3C and S5). Globally, 1,105 mRNA transcripts were differentially expressed by CORT infusion, CONS differentially regulated 990 compared to 419 by PULS, representing 89.6% and 37.9% of all transcripts respectively (S6A Fig). Additionally, 679 of these transcripts were differentially expressed by CONS only and no other infused condition, whilst PULS induced 109 (S6B Fig). The similarity between CONS regulation of total RNA-Seq data and the pSer2 Pol2 cohort is intriguing, as globally PULS was overly activatory (277 up- as opposed to 142 down-regulated) whilst CONS was equally activatory and inhibitory (497 up- to 493 down-regulated) of mRNA at 180 min despite the greater number of differentially expressed mRNA. Together, these data indicate transient increases, as opposed to decreases, in pSer2 Pol2 occupancy at PEAK_140 are more reflective of increased levels of mRNA at 180 min for PULS whilst CONS is a stronger transcriptional modulator for both up- and down-regulation.

The pSer2 Pol2 ChIP-seq data analysis may provide some insight, as despite PEAK_140 inducing the greater change in pSer2 Pol2 regulation, the changes are short-lived and highly transient, while the lower but sustained pSer2 Pol2 regulation by CONS has greater effects on mRNA expression levels, highlighting the complexity of regulation in this highly dynamic system.

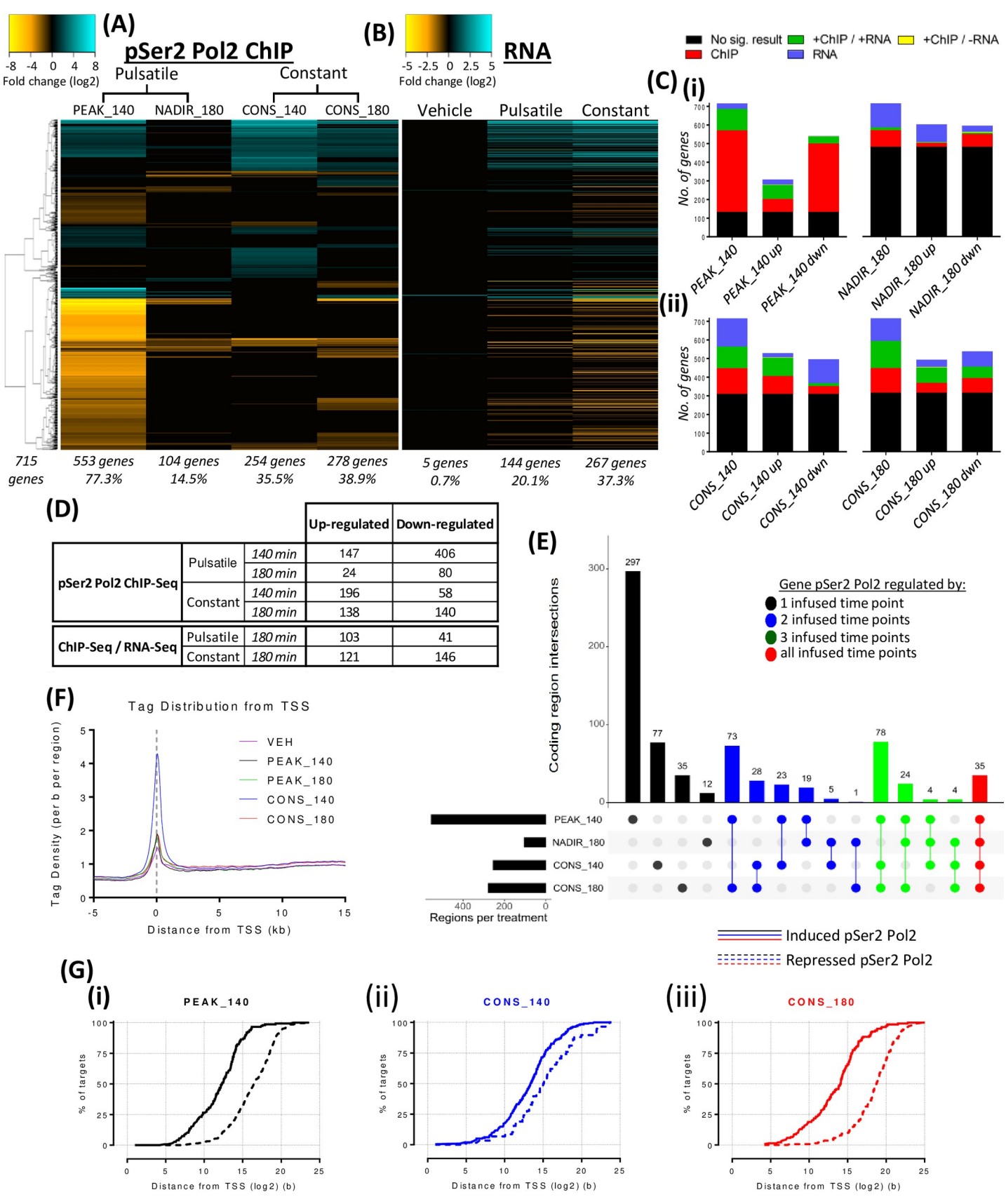

**Fig 3. Analysis of pSer2 Pol2 intragenic occupancy in response to PULS or CONS after 140 min and 180 min.** Heatmap visualises fold change in intragenic pSer2 Pol2 enrichment at CORT infusion-regulated genes in the liver in response to (A) PULS or CONS at 140 min and 180 min compared to VEH. (B) Fold changes of mRNA transcripts for genes identified by the pSer2 Pol2 analysis in response to pulsatile and constant VEH (left column), pulsatile VEH and CORT (middle) and constant VEH and CORT after 180 min of infusion. Heatmap indicates merged replicate tags (N = 2), data was hierarchically clustered according to fold change (log2) and colour intensity indicates degree of fold change as indicated by the scale bar between -8 to 8 (top left) for pSer2 Pol2 and between -5 to 5 (right of middle) for RNA. Total number of genes differentially regulated is indicated bottom left, number of genes in response to infused CORT time point under each lane as well as the percentage of total genes. (C) Bar graphs represent no. of genes with a significant fold change from the pSer2 Pol2 data (ChIP) and associated mRNA (RNA). Results were split into (i) pulsatile CORT infusion and (ii) constant CORT infusions at both 140 min and 180 min. Bars represent all results from a time point, and then split further into up-regulated and down-regulated results according to pSer2 Pol2 changes. Key indicates significant fold change for pSer2 Pol2 but not mRNA (red), increased or decreased for both pSer2 Pol2 and mRNA (green), opposing for pSer2 Pol2 and mRNA (yellow), mRNA only (blue) and no significant change for either pSer2 Pol2 or mRNA (black), as indicated by the key (top). (D) Table describing the number of significantly increased or decreased pSer2 Pol2 occupied and transcribed genes compared to VEH. (E) Graph represents the number of genes that were regulated in comparison to VEH by a single or combination of PULS and CONS time points. Y-axis of the vertical bar graph indicates the number of genes whilst the dots represent if a gene was enriched by a single (black), two (blue), three (green) or all (red) CORT infused time points. Horizontal bar graph indicates the number of genes enriched by each CORT infused time point. (F) Histogram of tag density distribution -5 kb and +15 kb from the TSS of pSer2 Pol2 differentially regulated genes on the sense strand within coding regions > 10 kb in response to a CORT infused time point. (G) Genes were split by either gain (solid) or loss (dotted) in pSer2 Pol2 occupancy and percentage of total were plotted against the distance to the most proximal GR binding site induced by (i) PEAK_140 (black), (ii) CONS_140 (blue) or (iii) CONS_180 (red) time point.

## RNAPol2 dynamics reveal evidence for a promoter-proximal pausing model as well as a relationship between increased or decreased occupancy predicated by the proximity of an enriched GR binding site

Tag density distribution from the TSS of sense strand coded regions > 10 kb, revealed distinct increased density between +35 b to +95 b from the TSS across all datasets irrespective of treatment or time point (Fig 3F). As the pSer2 Pol2 complex is associated with the actively transcribing complex throughout the coding region [31,32], this pattern of enrichment may represent stalled complexes, as proposed within the bimodal, promoter-pausing model of transcription [33,34].

Additionally, an interesting relationship was observed in our data, irrespective of the CORT infused condition (Fig 3G). When distance of the most proximal GR binding site to a CORT-regulated TSS was plotted against either increased or decreased pSer2 Pol2 occupied genes, the 25th percentile was closer (9.81 b (log2)) for increasingly compared to decreasingly occupied targets (14.29 b (log2)) in response to PEAK_140 (Fig 3G(i)). Similar relationships were observed throughout the percentiles as well as in response to either CONS time point (Fig 3G). It should be noted there were reduced differences in response to CONS_140 due to the overall increased transcriptional recruitment compared to other infused conditions (Fig 3G(ii)). This is consistent with similar mRNA data in A549 cells, however, mechanisms underpinning the observation are not yet understood [35].

## Functional pathway analysis reveals evidence for differential regulation of key metabolic pathways

Pathway analysis was performed using pSer2 Pol2 significant fold changes as a proxy for transcriptional change and according to published data, significant predictions of activation (z-score $\geq$ 2) or inhibition (z-score $\leq$ -2) were calculated. Trends (z-score 1–2) of transient activation during pulsatile infusion (synchronised to PEAK_140) but sustained during constant infusion (across both CONS_140 and CONS_180 time points) were evident for pathways regulating inflammation of liver, concentration of glucose, quantity of glycogen and carbohydrate (Fig 4A). Interestingly, there was a juxtaposition between necrosis and cell death of hepatoma cell lines, with proliferation of liver cells which was further compounded by the CONS only activation of liver growth (Fig 4B). Taken together, these data indicate that glucose and carbohydrate metabolism, as well as hepatocyte cell turn over were more strongly associated with CONS exposure compared to the same dose of PULS. This is most strongly observed within

the super-pathway of methionine degradation (Fig 4A), as it was significantly predicted to be activated at PEAK_140 (z-score 2.24), CONS_140 (z-score 2.45) and CONS_180 (z-score 2.45). This prolonged prediction of activation could induce greater degradation of methionine, compared to the transient activation in response to PULS.

Other trends of activation / inhibition were predicted for pathways regulating levels of serotonin, melatonin, monosaccharides, triacylglycerols, lipids and fatty acids (Fig 4C). In particular, significantly predicted inhibition of cholesterol biosynthesis was reported at the NADIR_180 (z-score -2) with an accompanied strong trend at PEAK-140 (z-score -1.89), whereas no prediction was made for CONS_140 and a reduced trend at CONS_180 (Z-score -1), indicating the potential for reduced cholesterol levels in response to PULS. Pathways of dyslipidemia and hyperlipidemia were also enriched for CORT-regulated genes, corroborating the known relationship between GCs and these conditions.

Addition to these powerful and novel insights into dynamic CORT regulated genomic transcriptional activity, we further investigated whether changes to the mRNA transcriptome were also detectable in related metabolic pathways. Gene ontology analysis of the 1,126 differentially expressed mRNA identified 59 pathways enriched by PULS regulated targets, whilst CONS regulated targets were enriched in a greatly increased total of 179 pathways (Fig 4D). CONS was observed to have a greater transcriptional effect as 38 out of the 59 pathways enriched for PULS regulated targets were also identified in response to CONS with a greater number of CONS regulated genes associated for each pathway (Fig 4E).

There was a good deal of cross-over between pSer2 Pol2 ChIP and RNA pathway analysis, despite the acute timeframe. Within RNA gene ontology results, pathways of gluconeogenesis, fat cell differentiation and sequestering of triglycerides were enriched for PULS regulated genes whilst triglyceride biosynthesis, liver development and metabolism of glucose, cholesterol, lipid and fatty acids were for CONS (Fig 4F). Genes within insulin, glucagon, triglyceride metabolism and circadian rhythm pathways were enriched for both CORT pattern regulated genes, with CONS regulating a greater number within each pathway.

Together, these analyses demonstrate how functional pathways implicated in metabolic pathology could be significantly affected by patterned CORT exposure within a relatively acute 3 hr period.

## Pathway analysis reveals key CORT pattern regulated genes

Based upon pathway analysis, several key genes were identified with robust CORT pattern-dependent pSer2 Pol2 changes in occupancy, as well as proximal GR binding within 50 Kb of their TSS. For example, pSer2 Pol2 occupancy of the key gluconeogenic target serine dehydrogenase (*Sds*) was transiently increased by PULS but sustained by CONS, as was GR enrichment at a site ~-5 kb from the TSS (Fig 4H) which was represented by an increased induction of mRNA expression by CONS compared to PULS (4.2- and 2.8-fold respectively). Oppositely, pSer2 Pol2 occupancy of glucose-6-phosphatase catalytic subunit (*G6pc*) was transiently decreased (S7B Fig), whilst no change to VEH was detected at solute carrier family 37 member 4 (*Slc37a4*) (S7A Fig). In response to CONS, no change was found for G6pc, but Slc37a4 was decreased at both CONS_140 and CONS_180 to VEH. Similar CORT pattern-regulated binding to Sds was detected -6 kb and -12 kb from the *G6pc* and at the TSS for *Slc37a4*. These targets are important factors within gluconeogenesis and present key evidence glucose production and efflux could be altered by patterned CORT exposure.

CONS was indicated to have similar effects at key lipogenic targets, potentially increasing lipogenesis as well as triglyceride formation. For example, LPIN1 oxidises fatty acids for triglyceride formation and induces adipocyte differentiation [36]. Changes in pSer2 Pol2

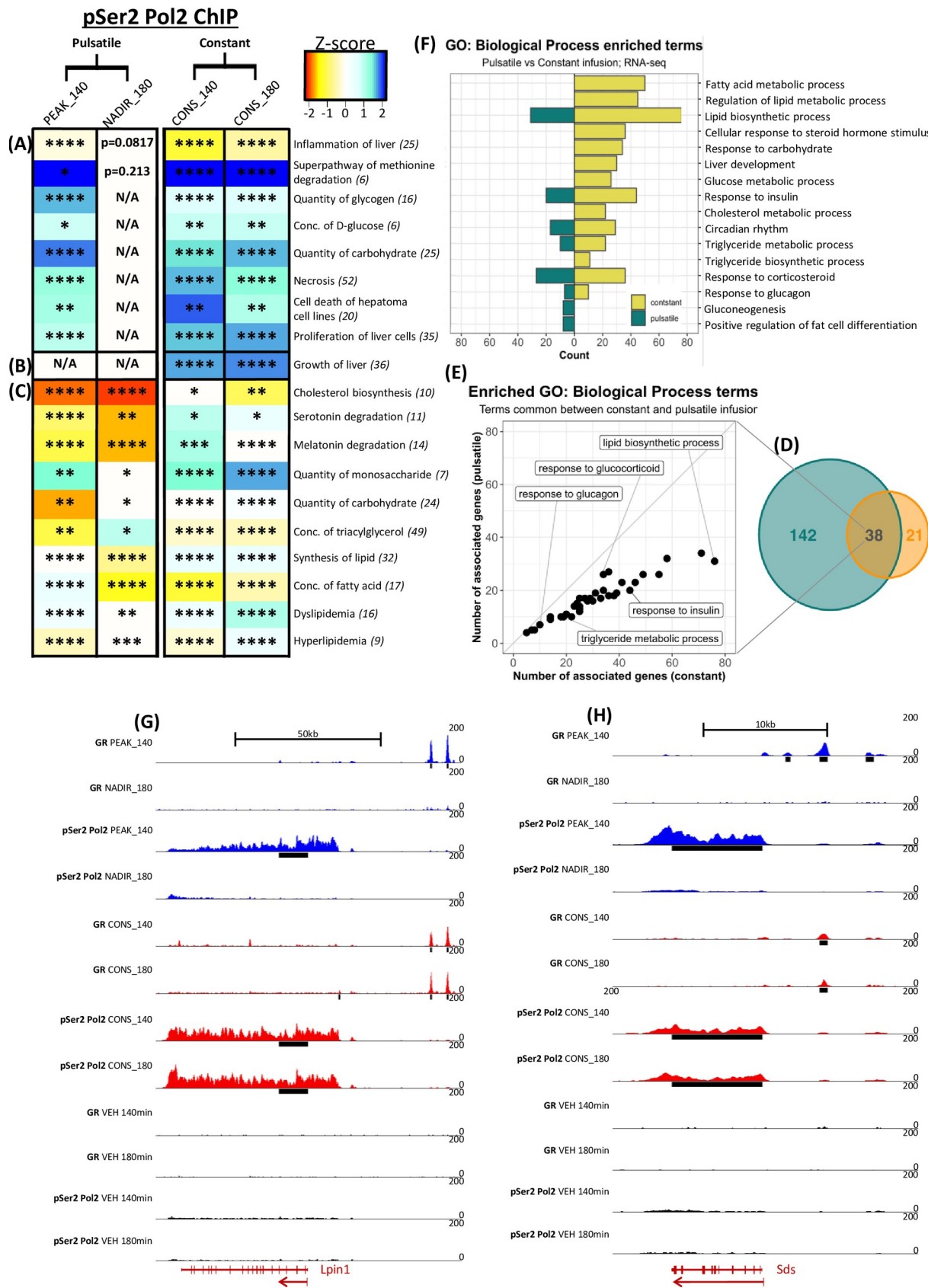

**Fig 4. Pathway analysis of differentially regulated genes reveal dynamically CORT-regulated metabolic pathways and targets.** (A-C) Heatmap list pathways identified as enriched for pSer2 Pol2 differentially occupied genes (total no. of genes indicated in brackets) in response to patterned CORT infusion at 140 min and 180 min compared to VEH. Pathways were sorted into (A) transiently enriched by PULS and sustained by CONS, (B) enriched by CONS only and (C) enriched at each time point for both infusion patterns VEH. Pathways enriched for significant changes in pSer2 Pol2 occupied genes, is indicated by p-value * < 0.05, ** < 0.01, *** < 0.001 and **** < 0.0001. Predicted activation of a pathway is indicated by a significant positive (≥ 2) z-score, whilst inhibition indicated by a negative (≤ -2) z-score as visualised by the colour intensity according to scale bar between -2 to 2. (D) Scaled Venn diagram representing the number of pathways identified with an over-representation of genes regulated by either pulsatile (orange), constant (blue) or both patterns of CORT compared to VEH from the RNA-Seq data set. (E) Within pathways enriched for pulsatile and constant CORT-pattern-regulated genes, number of genes within each pathway were plotted for either patterned CORT infusion. (F) Key metabolic pathways enriched for CORT regulated genes was selected and number of genes plotted in response to either pulsatile (left green) or constant (right yellow) infusions. Pathway analysis was applied to global RNA-Seq data and over-representation analysis calculated from genes with a log2 fold change > 0.585 and adjusted p-value < 0.05. (G,H) Both GR and pSer2 Pol2 tag density distribution was visualised using the UCSC genome browser within and extending from the TSS and 3' end of three GC-regulated targets in response to PULS (blue), CONS (red) and VEH (black) after 140 min and 180 min. (G) Three GR enriched sites -2 kb, -5 kb and -8.5 kb from the TSS of *Sds* were detected; the -5 kb of which was enriched at PEAK_140, CONS_140 and CONS_180 whilst the other two at PEAK_140 only. pSer2 Pol2 intragenic enrichment was increased in a similarly responsive manner. Data visualised over a chr12: 41,617,378–41,633,829 region. (H) Two similarly patterned enriched GR binding sites were detected -42 kb and -48 kb of the *Lpin1* TSS as was increased pSer2 Pol2 intragenic enrichment. Data visualised over a chr6: 41,789,357–41,899,861 region. Data visualised over a chr7: 102,581,386–102,596,166 region. All data was merged from two replicates and regions of CORT induced changes in enrichment to VEH are indicated by black bar below individual tracks. y-axis set to 0–200 and locations of introns and exons (maroon blocks) according to Ensembl rn6 co-ordinates are indicated as transcribed from the anti-sense strand (right to left–arrow) at the bottom.

indicated the transient increased transcription at PEAK_140, was sustained across time points by CONS (Fig 4G), and this is consequently associated with a greater fold-change in mRNA to VEH by CONS (3.5- and 4.4-fold increases respectively). Additionally, transient repression of apolipoprotein B (ApoB) which is integral to correct very low-density lipoprotein (VLDL) assembly, was sustained by CONS (S7C Fig). Together, CONS regulation of these targets could have distinct implication for metabolite deposition within the liver.

Transient increases / decreases in pSer2 Pol2 in response to PULS which are prolonged by CONS were also observed in key cell death and proliferative factors, such as Myc, Igfbp1 and Igf1 (S8A, S8B and S8C Fig). Most notably, dynamic increased transcription of Igfbp1 resulted in significant increased mRNA production in response to CONS only (3.1-fold); further evidence of prolonged changes in active transcription impacting overall levels of mRNA. Together these key targets indicate transient regulation of glucose and lipogenic processes, which exhibit the potential to be dysregulated during aberrant CORT exposure. Importantly, our results have also identified hepatocyte factors regulating turnover rates within the liver to be highly sensitive to the pattern of GC exposure.

## Discussion

For the first time we have been able to demonstrate, *in vivo*, that hourly PULS not only mimics physiological endogenous ultradian CORT fluctuations within the circulation of the rat but also robustly induces GR binding at over 2,500 binding sites synchronised to the peak of a CORT pulse (PEAK_140), whilst no significantly detectable binding events were found at NADIR_180. We also present evidence that resultant transcriptional activity is similarly dynamic, as the occupancy of over 500 genes by pSer2 Pol2 were similarly regulated and synchronised to the CORT pulse peak. These data demonstrate that GC-responsive mechanisms are sensitive to ultradian oscillations *in vivo* and reveal new evidence for differential transcriptional outcomes during pulsatile versus constant CORT replacement. For liver transcriptome regulation, pulsatile CORT induced GR binding and pSer2 Pol2 occupancy of GC-responsive genes was highly transient, whist in contrast constant CORT exposure induced prolonged GR binding and dysregulated RNAPol2 dynamic occupancy. This effect led to greater change in key mRNA targets in response to constant CORT, with potentially adverse impacts on key functional metabolic pathways in the liver. It was also striking that despite the greater changes

to pSer2 Pol2 occupancy at PEAK_140, CONS was a greater regulator of mRNA expression. Additionally, despite a greater repressive action at PEAK_140 on pSer2 Pol2 occupancy, it was the increased occupancy that was realised within transcriptomic changes in mRNA whilst CONS was equally effective for increased and repressed action. We therefore conclude that not only is the transcriptome sensitive to oscillating levels of endogenous glucocorticoids, but that the highly transient nature of PULS is not effective at repressing mRNA expression, which could play a potentially very important role for physiologically appropriate transcriptional regulation. Our pathway analysis highlighted several metabolic pathways enriched for CORT responsive targets regulating inflammation, glucose, lipid, carbohydrate, cholesterol regulation as well as liver growth and death pathways. Whilst these pathways are well known to be regulated by GCs [12,37], this is the first evidence for distinct and opposing transcriptional effects induced by different patterns of GC exposure. Notably, pulsatile CORT was significantly predicted to repress cholesterol biosynthesis at 180 min with a trend at 140 min, whilst constant CORT only predicted a reduced trend of repression at 180 min; a potential risk factor associated with hypertension, obesity, liver fibrosis and non-alcoholic steatohepatitis (NASH) [38–40]. Similarly, the impact of constant CORT exposure on prolonged activation of the methionine degradation pathway may be to increase the risk of fatty liver development; a condition associated with increased methionine metabolites within the liver [41,42]. Furthermore, dysregulation of ultradian CORT responsive genes could have serious adverse consequences for glucose, triglyceride and lipid metabolism, leading to increased risk of developing insulin resistance, type II diabetes and fatty liver [12,37,43,44], whilst dysregulating growth and apoptotic regulating genes as observed in with NASH and liver fibrotic phenotypes [45–47].

We have also identified CORT-regulated targets and associated GR binding sites which may have key roles in the development of metabolic syndrome phenotypes in gluconeogenesis, *de novo* lipogenesis, substrate production for triglyceride synthesis as well as lipoprotein function. For example, Sds deaminates serine into the metabolic substrate pyruvate [48], whilst *G6pc* and its associated carrier protein Slc37a4 convert glucose-6-phosphate into glucose in a rate limiting step of gluconeogenesis [49]. Despite changes in mRNA only detected for Sds, if these changes in genomic transcriptional activity were reflected in the transcriptome over a prolonged period, the result could be increased glucose production and efflux from hepatocytes. Additionally, ApoB is required for VLDL synthesis and function, facilitating lipid accumulation within the lipoprotein as well as endocytosis by intermediate density lipoproteins and lowering VLDL levels within the circulation [50,51]. As VLDL synthesis and function are integral to the livers ability to transport triglyceride, lipids and other metabolites, net loss of *ApoB* transcription over time in response to CONS could be fundamental, as loss of this protein could increase deposition and storage within the liver. Known risk factors of non-alcoholic fatty liver disease and steatosis [52–55]. Therefore, any dysregulation of synthesis over a prolonged period could have adverse effects.

Furthermore, dysregulating GR dynamics may also impact related endocrine-signalling systems. For example, growth hormone (GH) is secreted into the circulatory system in an inversed circadian fashion to CORT, peaking just after the onset of sleep [56,57]. The downstream targets of GH include stat5, stat3 and NFκB, which can act as transcriptional-co-factors with GR in well-established transrepressive mechanisms [58–60]. Therefore, dysregulation of GR dynamic activity could have serious implications for GH regulated pathways, particularly when one considers that night-shift workers present with unpredictable GH surges during their wake periods [57]. Increased bioavailability of active GR—also dysregulated by night-shift patterns—may then co-operatively increase the risk of metabolic syndrome associated clinical manifestations.

The importance of trying to reinstate a circadian rhythm in patients on GC replacement therapy is well recognised [61]. Surprisingly however, despite clinicians' best attempts to provide an 'optimal' replacement regime, many patients suffer from a range of metabolic and cognitive side effects [19,62]. The prevalence of side effects becomes worse with increasing dose, especially when using long-acting sGCs. A recent meta-analysis concluded prescription of dexamethasone and prednisolone were significant risk factors for development of insulin resistance within individuals with no underlying pathology [44] and corticosteroid treatment was shown to increase the risk of developing type II diabetes in a cohort of patients that did not present underlying risk factors [18]. Our data provide an ultradian chronobiological mechanism whereby phasic GR recruitment and dissociation from the DNA is needed to optimally regulate metabolic homeostatic pathways in the liver and a mechanism through which non-oscillatory activation of glucocorticoid receptors could increase risk of metabolic dysfunction.

Dysregulation of ultradian oscillations, which results in a glucocorticoid profile lacking the endogenous nadir phase, can increase the transcription of specific genes associated with deposition of triglyceride, cholesterol and carbohydrates within the liver. Coupled with reduced ApoB expression and correct VLDL synthesis, over a prolonged period the loss of ultradian oscillations could increase the risk of developing non-alcoholic fatty liver and steatosis phenotypes.

Previous strategies to reduce the side effect profiles of GCs have not considered the importance of the pattern of replacement or the difference in binding kinetics for different GC analogues. Since GC hormones are used in such a wide range of diseases and abnormalities of endogenous GC secretion have been implicated in many stress related and psychiatric conditions, we hope that our findings will contribute to a better appreciation of pulse pattern and lead to improvements in future therapeutics.

## Materials and methods

### Ethics statement

All animal procedures were carried out in accordance with the UK Home Office animal welfare regulations and approved by written consent from the Animal Welfare and Ethical Review Body (University of Bristol).

### Surgery and husbandry

Male Sprague-Dawley rats (250–300 g) (Harlan, Bicester, UK) were individually caged in soundproof rooms under standard conditions with standard chow and water available *ad libitum*. All CORT infusions were conducted in rats under 14:10 light/dark cycle (lights on at Zt 0 hr) whilst automated blood sampling experiments were under 12:12 light/dark cycle.

Rats were anaesthetised with a combination of Isoflurane (100% w/w liquid vapour (Merial, UK)) and medical air during bilateral adrenalectomy, jugular vein or/ and carotid artery cannulation. The cannula (Smith Medicals, UK) was exteriorised via craniotomy or a vascular access button and attached to a spring and swivel system (S9 Fig). Post-operative analgesic 1 mg of Carprofen (Rimadyl, Pfizer, UK), 25 µg dexamethasone (Sigma, UK) and 2.5 ml of glucose (5%)/ saline (0.9%) were administered subcutaneously. For 5 days post-surgical recovery, 0.9% saline drinking water was supplemented with 0.15 µg/L of corticosterone (Sigma, UK) until 16 hours prior to infusion, whereupon 0.9% saline drinking water was provided *ad libitum* to ensure complete CORT washout from circulation. Implanted cannula was 'flushed' daily via withdrawal of blood and replaced with fresh heparinised saline (1:100) to maintain patency.

## Circulating total blood serum CORT data

Total corticosterone blood serum samples were taken via an in-house automated blood sampler (S9 Fig). 40 μl blood samples were collected every 10 min in 160 μl Heparin: Saline solution (1:100). Plasma was separated from whole blood by centrifugation at 4000 rpm, 4˚C and diluted 1 in 50 with citrate buffer, processed in triplicate and incubated overnight in 50 μl of $^{125}$I corticosterone tracer (Oxford BioInnovation DSL Ltd, Oxford, UK) with 50 μl of rabbit anti-rat corticosterone antibody (kindly donated by G. Makara, Hungary). Free/ bound separation was performed using charcoal dextran precipitation and centrifuged pellets $^{125}$I corticosterone levels recorded using a gamma counter (Wizard-2470, Perkin Elmer, MA). Concentrations of corticosterone in each plasma sample were interpolated from a standard curve and intra- and inter-assay coefficient of variation have been established as 16.65% and 13.30%, respectively. Maximal corticosterone levels were identified by area under the curve analysis.

## Infusion profiles

To ensure complete clearance of any extra-adrenal endogenously produced corticosterone, all experiments were completed between Zt 13–16. *In vivo* 'mock' ultradian pulses were produced by a programmable pump (PHD Ultra Syringe Pump, Harvard Apparatus, USA) infusing CORT solution (solubilised 2-hydroxypropyl-β-cyclodextrin (HBC) complex carrier (Sigma, UK) in (1:100) heparinised, 0.9% saline solution) through the jugular vein at a dose of 3.84 μM. Each pulsatile infusion period began with a 20 min infusion at a rate of 1 ml per hour followed by cessation for 40 min and repeated hourly for 3 hrs. A sustained CORT-HBC saline solution was infused at a reduced rate of 0.33 ml per hour. To ensure all infusion lines and cannula were free of any air gaps that may have appeared between setup and infusion, the constant corticosterone infusion initial rate was increased to 1 ml per hour for 10 min before returning to 0.33 ml per hour. Area under the curve analysis indicated no difference in CORT dose delivered across the 3 hrs between the PULS (3,9087 ng/ ml X min ± 1,025 S.E.M.) and CONS (38,969 ng/ ml X min ± 1,007 S.E.M.). HBC dissolved in 0.9% saline solution and delivered in a pulsatile pattern was used as vehicle control infusion. Euthanasia time points corresponded to the pulsatile pattern, third corticosterone pulse peak and nadir at 140 min and 180 min respectively for all infusion patterns. Trunk blood samples were collected for each rat with 40 μl of 0.5 M ethylenediaminetetraacetic acid.

## Tissue collection

Liver was dissected and 0.4 g fixed in 1% (v/v) formaldehyde (Sigma, UK), phosphate buffered saline (1.37 M NaCl, 2.68 mM KCl, 10.14 mM $Na_2HPO_4$, pH 7.4) solution for 10 min at room temperature. Formaldehyde cross-linking was quenched with addition of glycine (final conc. 125 mM) for 5 min and washed three times in ice cold phosphate buffered saline supplemented with 2 mM NaF, 0.2 mM sodium orthovanadate and 1X cOmplete protease inhibitor (Roche Diagnostics). Fixed liver was stored at -80˚C in 0.2 g samples in 500 μl of S1 Buffer (10 mM HEPES, pH 7.9, 10 mM KCl, 15 mM $MgCl_2$, 0.1 mM (EDTA), pH 8) supplemented with 0.5 mM Dithiothreitol and 2 mM NaF, 0.2 mM sodium orthovanadate and 1X cOmplete protease inhibitor.

## Chromatin fragmentation

Samples were thawed slowly on ice and adjusted to a final volume of 1 ml/ sample with supplemented S1 buffer and Dounce homogenised. Lysate was centrifuged at 4000 rpm (4˚C) and

lysed in supplemented (2mM NaF and 0.2 mM sodium orthovanadate and 1X cOmplete protease inhibitor) sodium dodecyl sulphate (SDS) lysis buffer (2% SDS, 10 mM EDTA, 50 mM Tris-HCl (pH 8.1)). Chromatin was sonicated using a Branson Sonifier 450 (Branson Ultrasonics, Danbury, CT, USA) to 300–500 bp fragments with multiple 10-sec bursts at 10% output and centrifuged at 13,000 rpm at 4˚C to remove cellular debris from the chromatin suspension. Sheared chromatin was stored at -80˚C.

## ChIP assay

ChIP buffers were prepared in house, as described in the EZ ChIP kit protocol (Upstate Biotechnology, Lake Placid, NY, USA) with some modifications for use with animal tissue (as described where relevant). Sheared chromatin was removed from -80˚C storage and thawed slowly on ice, diluted to 50 μg in 100 μl of SDS lysis buffer and 0.9 ml with supplemented (2mM NaF, 0.2 mM NaVan and 1X cOmplete protease inhibitor) ChIP dilution buffer (0.01% SDS, 1.1% Triton X-100, 1.2 mM EDTA, 16.7 mM Tris-HCl pH 8.1, 167 mM NaCl). Inputs were immunoprecipitated against either a GR antibody cocktail (2 μg of PA1-510A, 4 μl of PA1511A (Thermo Fisher, USA) and 4 μg of M-20X sc-1004X (Santa Cruz, USA)) or the serine 2 phosphorylated RNA polymerase II complex (2 μl ab5095 (Abcam, UK)). Rabbit IgG antibodies were used as negative control (2 μg of sc-2027 (Santa Cruz, USA)). Inputs were incubated overnight at 4˚C and incubated with protein A Dynabeads for 4 hrs (Sigma-Aldrich, UK). To reduce non-specific binding, the DNA-Antibody-Dynabead slurries were sequentially washed with 150 mM salt buffer (0.1% SDS, 1% Triton X-100, 2 mM EDTA, 20 mM Tris-HCl pH 8.1), 500 mM salt buffer (0.1% SDS, 1% Triton X-100, 2 mM EDTA, 20 mM Tris-HCl pH 8.1), LiCl buffer (0.25 M LiCl, 1% IGEPAL-CA630, 1% deoxycholic acid sodium salt, 1 mM EDTA, 10 mM Tris-HCl pH 8.1) and twice in TE buffer (10 mM Tris-HCl, 1 mM EDTA) (pH 8.0). Complexes were eluted from the Dynabeads in 1% SDS 100 mM and 0.1M NaHCO3. NaCl was added (300 mM final concentration) and crosslinks reversed overnight at 65˚C. RNA was removed using RNase treatment (Roche Diagnostics) at 65˚C and protein was digested using proteinase K (Ambion, Huntington, UK) after adjusting each solution with EDTA (1 mM final) and Tris-HCl (4 mM final) at 45˚C. DNA was extracted using 25:24:1 phenol-chloroform-isoamyl alcohol (Sigma, UK) followed by 24:1 chloroform-isoamyl alcohol (Sigma, UK). DNA in the aqueous phase was precipitated overnight at -20˚C in 2.5 VOL 100% ethanol and 20 μg glycogen (Sigma-Aldrich, UK). DNA was pelleted by centrifugation at 13,000 rpm, 4˚C and washed in 70% Ethanol (13,000 rpm, 4˚C), air dried and suspended in 40 μl nuclease free water (Ambion, Huntington, UK).

## ChIP-Seq quality control and alignment

For each ChIP-seq replicate/condition/timepoint, liver samples from 6 rats were collected (72 rats in total). From each replicate/condition/timepoint, 10 x 100ug chromatin aliquots were prepared and immunoprecipitated with antibodies to either GR or pSer2 Pol2 before pooled, concentrated using a UniVapo Vacuum concentrator (Transcriptomics facility, UOB) to approx. 35 μl and sequenced. Note: Two pooled ChIP samples (N = 10 in each) were sequenced to allow for concordant peak calling analysis as previously described [63]. Library preparation (TrueSeq SBS v3-HS kit (illumina, US)) and Next-Gen sequencing using the Illumnina HiSeq2000 platform (Illumina, USA) of the samples was processed at the Advanced Technology Research Facility (ATRF), National Cancer Institute (NCI-Frederick, MD, USA).

Adapters were removed, filtered for a base quality of 33 and sequence lengths of 50 b were isolated using Trimmomatic-0.36 [64]. Trimmed FASTQ files were aligned to the rat genome

(UCSC rn6 assembly) using bowtie2 [65] and duplicate tags removed using SAMtools 1.3.1 [66]. Subsequent analysis was performed using HOMER v4.9 [67].

Counts were normalised to 10 million tags (makeTagDirectory) to allow for cross sample comparisons and visualised using the UCSC genome browser [68] (http://genome.ucsc.edu/).

### Identification of GR ChIP-Seq enriched regions

Enrichment of GR tags to 1% input control, were identified (findPeaks) using relaxed settings (-F1 -L1 -P.1 -LP.01 -poisson .1 -style factor). Replicate concordance was assessed using an irreproducible discovery rate (IDR v2.0.3) set to 0.01. Overlapping enrichments between replicates were merged using mergePeaks (HOMER v4.9.1) and filtered according to the IDR estimated confidence threshold. Overlapping confident enrichment regions were merged again (mergePeaks) to create a single list of enriched GR locations across conditions. Visualisation of tag densities were done using EASeq v1.101 [69] and gplots (heatmap.2).

### Analysis of differential enrichments

Analyses of pSer2 Pol2 ChIP-Seq enrichment was restricted to transcript coding regions larger than twice the fragment length ($>$ 320b) and limited to 10 Kb from the TSS (Ensemble release Rnor_6.0.92). Raw input ChIP-Seq tag counts were subtracted from both GR and pSer2 Pol enrichment regions (annotatePeaks.pl–HOMER v4.9.1).

Differential GR and pSer2 Pol2 enrichment fold changes were assessed to VEH control and between corticosterone infusion time points using getDiffExpression.pl (HOMER v4.9.1) and DESeq2 [70]. GR tags were normalised to the count within *de novo* binding regions whilst pSer2 Pol2 tags were normalised to total sequenced tags across replicates. Differential GR and pSer2 Pol2 enrichments were considered significantly different if fold change $>$ 1.5-fold and FDR $<$ 0.05. All other results were given the value 0. Variants of the same gene were filtered for the greatest fold change across all conditions and time points. Distances between TSSs and GR binding regions were assessed using annotatePeaks.pl.

### Motif analysis

223 b in either direction from the centre of identified GR enrichment regions (total equal to twice the average fragment length) were analysed for both *de novo* and known motifs using findMotifsGenome.pl (HOMER v4.9.1). Repeat sequences were masked and optimised for motifs 8, 10 and 12 b long. Calculations of motifs per base pair per peaks were calculated using de novo motif matrices within HOMER v4.9.1 and histograms indicate location of the motif relative to the GR binding site centre and plotted using GraphPad Prism v6.07 for Windows (La Jolla California, USA, www.graphpad.com).

### RNA preparation and sequencing

Frozen liver sections were homogenised (Polytron, PT-2100, Kinematica, CH) in TRIzol (ThermoFisher Scientific, USA). The aqueous phase was separated via chloroform extraction (Sigma-Aldritch, UK) and centrifugation (10,000 rpm, 4˚C), total RNA extracted, and DNA digested (miRNAeasy mini, Qiagen, US). Libraries were constructed from 1 µg total RNA (TrueSeq Stranded Total RNA kit, Illumina, USA) and filtered for a RIN score $>$ 8 and $<$ 10 (Agilent Tape Station, Bioanalyzer, UK) for sequencing (HiSeq 2500, Illumina UK). All samples were handled according to manufacturer protocols. Adapters removed, sequences trimmed (bbduk.sh) and FASTQ files aligned to the rat genome (UCSC rn6 assembly).

### RNA-Seq analysis

Exonic co-ordinates according to the Ensemble, Rnor_6.0.92 release were assessed (feature-Counts) [71], filtered $> 10$ reads and differential expression of transcripts with a fold change $> 1.5$ and FDR $< 0.05$ assessed via DESeq2 [70]. Visualisation of results were plotted using gplots (heatmap.2).

### pSer2 Pol2 pathway analysis

pSer2 Pol2 ChIP-Seq DESeq2 data were analysed via Ingenuity Pathway Analysis software (Qiagen Inc., https://www.qiagenbioinformatics.com/products/igenuity-pathway-analysis). Enrichment of pathways were identified from genes with fold changes $\geq 1.5$-fold and adjusted p-value $< 0.05$ to VEH based on previously published liver tissue and HepG3, hepatoma, hepatocyte cell lines as well as mice, rat and human data. Positive and negative z-scores represent either predicted activation ($\geq 2$) or inhibition ($\leq -2$) respectively of pathways are based upon the degree of fold change and significance of the individual genes change in expression, the numbers of genes that act as an upstream regulator to induce and / or inhibit a specific pathway, adjusted for the number of studies that report this effect, and according to a binomial distribution. As explained by the following equation, $z = \frac{x}{\sigma_x} = \frac{\sum_i w_i x_i}{\sqrt{\sum_i w_i^2}}$. Where the distribution of $x = \sum_i x_i = N_+ + N_-$, the variance of $\sigma_x^2 = N\sigma^2 = N$ and the adjustment of published findings is $w_i = \frac{|M_{activating} - M_{inhibting}|}{M_{activating} + M_{inhibting} + 1}$.

### RNA pathway analysis

ENSEMBL [72] was used to retrieve genome annotations using biomaRt (v2.44.4) [73,74], org. Rn.eg.db (v3.11.4), AnnotationDbi (v1.50.3) packages in R. Scaled Venn diagrams were generated using VennDiagram (v1.6.20) package in R [75]. Gene-ontology [76,77] analysis was performed using ClusterProfiler (v3.16.1) package in R [78,79]. Over-representation analysis was applied for genes passing adjusted p-value $< 0.05$ and log2 fold-change $> 0.585$. All expressed genes were used as the background in the over-representation analysis and as the input gene list in the gene-set enrichment analysis [80]. Benjamini-Hochberg correction (adjusted p-value $< 0.05$) was used for multiple comparison correction in the pathway analysis. Pathway analysis results were visualised using custom scripts written in R [81].

### Statistical information

Regular two-way ANOVA was used to test for significant pattern, time-dependent and interactions followed by Bonferroni post-hoc testing ($p < 0.05$). All data is expressed as mean ± s.e.m. where appropriate.

## Supporting information

**S1 Fig. 24 hour circulating CORT profiles in individual male Sprague Dawley rats.** (A-E) Blood samples were collected every 10 min over a 24-hour period from rats and total CORT levels measured in the plasma by radioimmunological assay. Light schedule was 12:12 (on Zt 0 / off Zt 12) with dark phase indicated by shaded bar along the top of the plot. (TIF)

**S2 Fig. Sequenced GR tag distribution across identified binding regions.** (A-E) Tag density at identified GR enriched regions across conditions was plotted across a series of histograms in response to (A) VEH, (B) PEAK_140, (C) NADIR_180, (D) CONS_140 and (E) CONS_180.

(F-J) Histograms of tag density were also plotted at sites with reduced enrichment in response to CORT infusion compared to VEH. In all cases, for (F) VEH, (G) PEAK_140, (H) NADIR_180, (I) CONS_140 and (J) CONS_180, robust peaks in tag density were localised at enrichment centre and reduced to negligible levels within 1 kb in either direction. Raw tags (log2) of each replicate (N = 2) were segmented in 5 b bins, spanning 5 kb in each direction from enrichment centre.
(TIF)

**S3 Fig. Heatmaps of GR and pSer2 Pol2 enrichment in response to a single or combination of patterned CORT infusion time points.** Results visualised within these heatmaps were assessed within differentially enriched regions to VEH in response to the indicated pulsatile or constant CORT infused time point. (A) GR time dependent sites were defined as sites enriched to VEH in response to pulsatile or constant CORT, that were also significantly regulated between 140 min and 180 min CORT infused time points. Of those, fold change is visualised for 140 min relative to 180 min time points for both pulsatile and constant CORT infusion. (B and C) GR sites identified as differentially enriched relative to VEH were assessed in significant fold change comparisons, with (B) PEAK_140 fold changes in enrichment visualised in comparison to CONS_140 and CONS_180 and (C) NADIR_180 fold changes in enrichment visualised in comparison to CONS_140 and CONS_180. (D) pSer2 Pol2 time dependent sites were defined as sites that had been identified as significantly differentially enriched relative to VEH, then found to have a significantly difference between 140 min and 180 min, with fold change in 140 min relative to 180 min visualised here for both pulsatile or constant CORT infusion. (E and F) pSer2 Pol2 sites differentially enriched to VEH by (E) PEAK_140 and (F) NADIR_180. In both cases, fold changes in enrichment are visualised at these sites compared to CONS_140 and CONS_180. Heatmap indicates merged replicate tags (N = 2), colour intensity indicates degree of fold change compared to VEH as shown within the -8 to 8 scale bar and regions of interest clustered by fold changes hierarchically clustered by Euclidean distances.
(TIF)

**S4 Fig. Distribution of motifs within GR enriched regions.** (A) Bar graph displays the top 5 most significant motifs discovered within GR bound, GRE containing regions across all CORT infused time-points. (B) Motifs identified in GR binding regions that were enriched by PEAK_140 (white), CONS_140 (grey) or CONS_180 (black) time points compared to VEH. (C) Lanes 1–4 of heatmap indicates fold change in GR enrichment within identified binding regions in the liver in response to either PLS or CONS at 140 min and 180 min compared to VEH. Data presents merged replicate tags (N = 2) hierarchically clustered according to fold change (log2) and colour intensity indicates degree of fold change as indicated by the scale bar between -8 to 8 (top left). Lanes 5–10 indicate the presence (blue) or not (black) of the motif indicated at the top of the lane within the GR enriched site (horizontal row of heatmap). *De novo* motifs were identified within 233 b regions in either direction from the centre of identified GR enriched regions.
(TIF)

**S5 Fig. Table of differentially occupied pSer2 pol2 genes at 140 min and 180 min and associated differentially expressed RNA results at 180 min in response to pulsatile or constant CORT infusion and compared to VEH.** Of the 715 differentially pSer2 Pol2 occupied genes; results in the top row (row 1) are split into differentially occupied pSer2 Pol2 ChIP only (red), as well as genes that were differentially occupied and expressed within the RNA-Seq data set (green) across all CORT infused conditions. Rows 2–5 split data by 140 min and 180 min time

points in response to pulsatile or constant CORT infusion and whether the increased or decreased fold change was observed in both data sets (green) or were opposingly increased / decreased (yellow), the number of non-significant ChIP results (black) and genes whose RNA was differentially expressed without an associated significant ChIP result (blue). Rows 6–9 split data by up-regulated and down-regulated pSer2 Pol2 occupation at PEAK_140 and NADIR_140 and RNA expression at 180 min whilst rows 10–13 split data for constant CORT infusion in the same manner. Data represents fold changes > 1.5 and adjusted p-value < 0.05. (TIF)

**S6 Fig. Differentially expressed RNA in response to 180 min pulsatile, constant or vehicle infusion.** (A) Fold changes of 1,105 differentially expressed exonic RNA transcripts (bottom left) was visualised for pulsatile vehicle compared to constant vehicle (left), pulsatile CORT compared to pulsatile vehicle (middle) and constant CORT compared to constant vehicle infusion (right). Heatmap indicates merged replicates (N = 3), colour intensity indicates degree of fold change compared to VEH as shown within the -6 to 6 scale bar and genes clustered by log2 fold change values, hierarchically clustered by euclidean distances as indicated by dendrogram (far left). Number of genes differentially regulated are indicated under the appropriate lane. (B) VENN diagram indicates the number of genes differentially expressed by a single or combination of infused pulsatile / constant CORT or vehicle comparisons, as observed within the heatmap (A). (TIF)

**S7 Fig. GR and pSer2 Pol2 enrichment in response to patterned CORT infusion at a selection of key metabolic targets.** Both GR and pSer2 Pol2 tag density distribution was visualised using the UCSC genome browser within and extending from the TSS and 3' end of three GC-regulated targets in response to PULS (blue), CONS (red) and VEH (black) after 140 min and 180 min. (A) GR enrichment was identified at the TSS of *Slc37a4* in response to PLS_140 and CONS time points whilst pSer2 Pol2 intragenic enrichment increased in response to CONS only. Data visualised over a chr8: 48,716,099–48,725,227 region and y-axis set to 200. (B) Similarly, responsive GR enrichment was also detected 6 kb upstream of the *G6pc* TSS but pSer2 Pol2 intragenic enrichment was lost in response to PEAK_140 only. Data visualised over a chr10: 89,275,804–89,306,418 region and y-axis set to 200. (C) GR enrichment was identified in response to PEAK_140 only, 26 kb upstream of the *ApoB* TSS, whilst intragenic pSer2 Pol2 enrichment was reduced at PEAK_140, CONS_140 and CONS_180 for *ApoB*. Data visualised over a chr6: 33,145,813–33,224,922 region and y-axis set to 200. Data was merged from two replicates, normalised to 10 million tags and visualised using the UCSC genome browser within and extending from the TSS and 3' end. Regions of CORT induced changes in enrichment to VEH are indicated by black bar below individual tracks. Scale bar is indicated at the top and locations of introns and exons (maroon blocks) according to Ensembl rn6 co-ordinates are indicated as transcribed from either the sense (left to right) or anti-sense strand (right to left) according to arrow direction (bottom). (TIF)

**S8 Fig. GR and pSer2 Pol2 enrichment in response to patterned CORT infusion at a selection of key metabolic targets.** Both GR and pSer2 Pol2 tag density distribution was visualised using the UCSC genome browser within and extending from the TSS and 3' end of three GC-regulated targets in response to PULS (blue), CONS (red) and VEH (black) after 140 min and 180 min. (A) No enriched GR sites were detected 50 kb up- or downstream of the *Myc* TSS, but intragenic pSer2 Pol2 enrichment was increased at PEAK_140, CONS_140 and CONS_180. Data visualised over a chr7: 102,581,386–102,596,166 region and y-axis set to 200.

(B) CORT responsive GR enrichment was detected 5 kb upstream of the *Igfbp1* TSS as was pSer2 Pol2 intragenic enrichment. Data visualised over a chr14: 87,441,155–87,456,358 and y-axis set to 200. (C) GR enrichment was detected 37.5 kb downstream within an intronic region of *Igf1* in response to PEAK_140, CONS_140 and CONS_180 as was increased pSer2 Pol2 intragenic enrichment. Data visualised over a chr7: 28,393,595–28,505,212 region and y-axis for GR set to 200 and pSer2 Pol2 to 100. Data was merged from two replicates, normalised to 10 million tags and regions of CORT induced changes in enrichment to VEH are indicated by black bar below individual tracks. Scale bar is indicated at the top, y-axis set to 0–200 and locations of introns and exons (maroon blocks) according to Ensembl rn6 co-ordinates are indicated as transcribed from sense (left to right) or the anti-sense (right to left) strand as indicated by the arrow (bottom).
(TIF)

**S9 Fig. Schematic of programmable pump and automated blood sampling system, related to Methods, Surgery and Husbandry.** Rats implanted with intravenous cannulae were exteriorised either via craniotomy or vascular access "button" over the shoulders and attached to swivel to allow free movement. Solubilised CORT (blue) was infused via programmable pump (left). Automatic blood sampling system (right) withdrew blood (red) from the intravenous cannula through the swivel and valve/ switch system powered by a reversible pump. Once the required volume has been withdrawn, the pump is reversed, and the valves direct the flow to infuse the same volume of heparinised saline solution (black–top right) back through the intravenous cannula. Arrows denote direction of travel.
(TIF)

**S1 Data. Raw DESeq2 results from GR ChIP-Seq experiments.** Raw output from DESeq2 (HOMER, getDiffExpression.pl) analysis of GR read counts in response to CORT infused treatments, compared to VEH at *de novo* enriched regions to 1% input control. Spreadsheet detail's chromosomal locations of enrichments (columns A-E) and whether they were 'Enriched' for GR tags compared to 1% input control in response to each CORT and VEH infused time point (F-J), genetic feature information (M-Z), raw tag counts for each replicate (AA-AJ) and DESeq2 results including log2 fold changes and associated p-values (columns AK-BW).
(XLSX)

**S2 Data. DESeq2 results from GR ChIP-Seq experiments, filtered for significant enrichment by CORT infusion compared to VEH.** Raw output from DESeq2 (HOMER, getDiffExpression.pl) of GR read counts in response to CORT infused treatments and compared to VEH at *de novo* GR enriched regions to 1% input control. Results have been filtered for log2 fold change > 0.585 or < -0.585 and adjusted p-value < 0.05. Any results that did not meet these criteria were removed from the data or assigned 0 values in cases where one or more CORT infused treatments were significantly enriched. Spreadsheet detail's chromosomal locations of enrichments (columns A-E) and whether they were 'Enriched' for GR tags compared to 1% input control in response to each CORT and VEH infused time point (F-J), genetic feature information (M-X), raw tag counts for each replicate (Y-AH) and DESeq2 results including log2 fold changes and associated p-values (columns AI-BW).
(XLSX)

**S3 Data. GR ChIP-Seq DESeq2 results used to generate heatmaps.** First 5 columns indicate log2 fold change in GR enrichment compared to VEH after PEAK_140, NADIR_180, CONS_140 and CONS_180 to VEH (columns 2–5). Further six columns indicate if one / multiple of the top 6 most significant *de novo* over-represented consensus sequences were found

within the enrichment region.
(CSV)

**S4 Data. DESeq2 results from pSer2 Pol2 ChIP-Seq experiments, filtered for significant enrichment by CORT infusion compared to VEH.** Raw output from DESeq2 (HOMER, get-DiffExpression.pl) analysis of pSer2 Pol2 read counts in response to CORT infused treatments and compared to VEH within transcript coding regions restricted to larger than twice the fragment length ($>$ 320b) and limited to 10 Kb from the TSS. Results were filtered for log2 fold change $>$ 0.585 and adjusted p-value $<$ 0.05 in enrichment. Any results that did not meet these criteria for any infused time point were removed from the data. Spreadsheet columns detail's Ensembl ID (A), gene name (B), enrichment region (C-F), genetic feature information (G-I), raw tag counts for each replicate(J-S) and log2 fold changes and associated p-values (T-AW).
(CSV)

**S5 Data. Analysis of pSer2 Pol2 occupation DESeq2 results by Ingenuity Pathway Analysis.** Within tab "Raw data", describes DESeq2 fold change and p-value results used for Ingenuity Pathway Analysis. Columns describe Ensemble gene ID (A, B) and gene name (C) as well as log fold change, log2 fold change, p-value and adjusted p-value for PEAK_140 (D-G), NADIR_180 (H-K), CONS_140 (L-O) and CONS_140 (P-S) comparisons to VEH. Within tab "IPA results–pathway Z scores", describes Ingenuity Pathway Analysis z-scores for each pathway. Columns describe pathway (A) and the z-score in response to PEAK_140 (B), NADIR_180 (C), CONS_140 (D) and CONS_180 (E) compared to VEH. Within tab "Heatmap Z scores", describes the results used to plot Fig 4A, 4B and 4C. Columns describe pathway (A) and the z-score in response to PEAK_140 (B), NADIR_180 (C), CONS_140 (D) and CONS_180 (E) in comparison to VEH. Within tab "IPA results–sig. gene", describes differentially enriched genes known to either 'activate' or 'inhibit' pathways visualised in Fig 4A–4C. Columns describe pathway (A), the differentially regulated gene name (B) and the log2 fold change in response to PEAK_140 (C), NADIR_180 (D), CONS_140 (E) and CONS_180 (F) in comparison to VEH. Genes are labelled as either known to be activatory (Act) or inhibitory (Inh) (G) within that pathway.
(XLSX)

**S6 Data. Raw DESeq2 results from RNA-Seq experiments in response to CORT infusion compared to VEH.** Raw, normalised count output from DESeq2 (HOMER, getDiffExpression.pl) of exonic RNA-Seq count data in response to CORT infused treatments compared to VEH. Spreadsheet detail's Ensembl ID followed by the baseMEAN (average normalised count values divided by the size of factors over all samples), log2 fold change values, the standard error and both p-value and adjusted p-value of the observation for VEH, PULS and CONS.
(CSV)

**S7 Data. Filtered DESeq2 results for significant fold change in enrichment of mRNA in response to pulsatile or constant CORT infusion compared to VEH.** Raw, normalised count output from DESeq2 (R) of exonic RNA-Seq count data in response to CORT infused treatments and compared to VEH. Results were filtered for log2 fold change $>$ 0.585 or $<$ -0.585 and adjusted p-value $<$ 0.05 in enrichment. Any results that did not meet these criteria were removed from the data or assigned 0 values in cases where 1 or more CORT infused treatments were significantly occupied. Spreadsheet columns detail's Ensembl ID and gene name (A, B) followed by the baseMEAN (average normalised count values divided by the size of factors over all samples), log2 fold change value and adjusted p-value of the observation for VEH (C,

D), PULS (E, F) and CONS (G, H).
(CSV)

**S8 Data. Pathway analysis of differentially regulated RNA-Seq genes in response to CORT infusion compared to VEH.** Gene Ontology, KEGG, Reactome and WIKI pathway analysis of unfiltered RNA-Seq DESeq2 data in response to pulsatile or constant CORT infusion compared to pulsatile or constant VEH respectively. Results for pulsatile and constant infusion are on separate sheets, and each spreadsheet details process ID, brief description of pathway and rank (BD) as well further detailed information (E-J) for each analysis.
(XLSX)

## Author Contributions

**Conceptualization:** Stafford L. Lightman, Becky L. Conway-Campbell.

**Data curation:** Benjamin P. Flynn.

**Formal analysis:** Benjamin P. Flynn, Audrys G. Pauza, Sohyoung Kim, Songjoon Baek, Mark F. Rogers, Alex R. Paterson.

**Funding acquisition:** Stafford L. Lightman, Becky L. Conway-Campbell.

**Investigation:** Benjamin P. Flynn, Matthew T. Birnie, Yvonne M. Kershaw, Becky L. Conway-Campbell.

**Methodology:** Benjamin P. Flynn, Matthew T. Birnie, Yvonne M. Kershaw, Becky L. Conway-Campbell.

**Resources:** Diana A. Stavreva, David Murphy, Gordon L. Hager.

**Writing – original draft:** Benjamin P. Flynn.

**Writing – review & editing:** Benjamin P. Flynn, Stafford L. Lightman, Becky L. Conway-Campbell.

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
