## [Decision Letter · Decision Letter 0]

21 Aug 2020

Dear Dr Flynn,

Thank you very much for submitting your Research Article entitled 'Corticosterone pattern-dependent glucocorticoid receptor binding and transcriptional regulation within the liver' to PLOS Genetics. Your manuscript was fully evaluated at the editorial level and by independent peer reviewers. The reviewers appreciated the attention to an important problem, but raised some substantial concerns about the current manuscript. Based on the reviews, we will not be able to accept this version of the manuscript, but we would be willing to review again a much-revised version. We cannot, of course, promise publication at that time.

**Comments to the Authors:**

Reviewer #1: The manuscript by Flynn and collogues builds on a number of studies from this lab seeking to characterize and understand the function of ultradian corticosterone/cortisol secretion in rodents and humans. The lab has published numerous studies showing that ultradian cort secretion manifests in dynamic GR occupancy, chromatin remodeling and gene expression. Most of these studies have been performed in cell cultures and little evidence is available from in vivo based model systems. In this manuscript Flynn et al uses a rodent model system to induce ultradian cort levels in serum and subsequently study genome-wide occupancy of GR in liver tissue. This is combined with genome-wide occupancy of elongating RNA polymerase II to probe for gene regulatory effects. Importantly, in agreement with previously published in vitro based data this study shows dynamic GR occupancy in response to pulsatile cort treatment. Interestingly, some regions of the genome show less pronounced GR occupancy in constant cort treatment compared to pulsatile cort treatment. Also, RNA polymerase II occupancy suggests that the impact on liver gene transcription may be differentially regulated in pulsatile versus constant cort treatment.

This study is of great importance in the efforts to understand the function of ultradian corticosterone/cortisol secretion in vivo. However, although the experimental setup is sound, the data analysis could be improved (see below). Also, the evaluation of gene expression using RNA polymerase occupancy could be improved by supported RNAseq analysis. Moreover, the study lacks mechanistic data or even suggestions to mechanisms implicated in constant versus pulsatile impact on GR occupancy and gene regulation. Suggested mechanisms may be more evident with additional data and data analysis.

Major comments:

Experimental setup. The authors use a very elegant experimental setup to mimic ultradian serum cort levels in a rat. As stated in the experimental procedures cort was given to adrenalectomized rats for 20min followed by a 40 min pause. This cort pulse was repeated three times over 3 hours and livers were collected at cort pulse peaks and nadir. However, it is unclear when during the day this cort pulse or constant cort was administered. As the authors nicely show in figure 1A-C ultradian cort secretion is most pronounced at the light/dark transition during a circadian rhythm. Thus, to mimic ultradian cort secretion and study the impact on gene expression it is also important to consider the time of the experiment. Especially since GR response to cort and subsequent regulation of gene expression is highly dependent on when cort is administered (PMID: 30179226). The authors should indicate when during a circadian rhythm the cort pulses were induced and why this particular timepoint was chosen.

ChIP-seq against GR in constant versus pulsatile cort conditions demonstrates a clear difference in GR occupancy under these two different conditions. How does this compare to a pool of GR ChIP-seq experiment using livers isolated around ZT12 during a circadian rhythm? In other words, how does the identified GR binding sites in the pulsatile/constant cort treatment compare with a normal physiological situation?

The clustering shown in figure 2G suggests that GR occupancy is reduced in constant 140min cort compared to 140min pulsatile cort at about a half of the GR binding sites. Have the authors performed separate de novo motif analysis of these seemingly pulsatile specific GR binding sites and GR binding sites not occupied by GR under constant cort treatment? Also have the authors compared the GRE and half-GRE (referred to as AR half-site) motif score of these two types of GR binding sites?

GR occupancy of chromatin does not necessarily lead differential regulation of enhancer activity. To determine if pulsatile and constant cort leads to difference in enhancer activity the authors should measure H3K27Ac at the identified GR binding sites. Alternatively, they could quantify RNA polymerase II at these sites as a proxy of enhancer RNA synthesis which correlates well with differential enhancer activity. Or total RNA-seq could provide insights to synthesis of eRNA.

In figure 3 the authors show results from quantification of RNA polymerase II ser2 phosphorylation ChIP-seq. The quantification is performed at intragenic regions of the genome. Thus, referring to genes. They refer to these genes as sites or regions in figure 3 (panels A and F). Would be more appropriate with a consistent reference to genes instead. A consistent reference to genes may also be more appropriate in the text describing the figure. Here the intragenic regions are referred to as region, sites or genes. Would be most informative to consistently call these regions genes.

It is clear from figure 3A that the difference between pulse and constant cort is predominantly observed for repressed genes, while induced genes seems more alike. Yet, the comparison in figure 3F does not discriminate between induced and repressed genes. It would be informative to split figure 3F into induced and repressed genes.

Unclear what is plotted in figures 3B-D and what information this specifically provides.

Induced genes by cort pulse (140 peak) and constant cort (140 cons) seems similar. What is the log2FC pearson correlation between these two conditions for induced and repressed genes? Could be illustrated by a scatterplot.

It is very interesting that pulsed cort treatment leads to reduced elongating pol2 at genes while constant cort treatment leads to little or no reduction of pol2 occupancy. Is this reflected by RNA expression levels? In other words, does RNAseq using total RNA (e.g. ribo-zero prepared cDNA libraries) as input followed by intron quantification lead to the same conclusion? If RNA expression is reduced in both conditions this may indicate a different mechanism behind gene repression by pulsed and constant cort treatment.

In figure 3G pol2ser5-P distribution is plotted for all differentially regulated genes. Again, it would be informative to plot this separately for upregulated and downregulated genes. Maybe the pronounced peak shown in the CONS_140 condition is a result of stalled pol2 in that condition. Comparing with RNAseq data (mentioned above) could indicate if genes may be repressed as a consequence of stalled pol2.

It would be informative to also split genes/TSS plotted in figure 3H into induced and repressed genes. For example, figure 3Hi have an overrepresentation of repressed genes compared to genes/TSS plotted in figure 3Hii. This may explain the differences seen.

Figure 4A lacks information on the number of genes identified in each pathway. A dotplot like approach would provide this information together with a p value. Unclear how the z-score is calculated, and how this informs on negative and positive regulation of the pathway.

Minor comments:

Occupation is used frequently to refer to GR and pol2 binding to chromatin. Occupancy would be a more appropriate word.

Genome browser data shown in figures 4, S4 and S5 lacks labeling of the y-axis.

Reviewer #2: Flynn et al use an adrenalectionised rat model with cortisol (CORT) replacement in either a pulsatile plasma or continuous replacement pattern to assess the plasma pattern-dependent effects of CORT on gene expression in liver. The results confirm many earlier studies, by the present authors and others, in various cell culture models showing CORT pattern-dependent effects on genomics, epigenetics and gene expression (refs# 14, 25-28). Further, similar (albeit less comprehensive) CORT pulse-dependent effects were reported by the authors in an adrenalectomized rat model back in 2009 using a high CORT dose (Ref #14). The present study adds new information, most notably mapping to pattern-dependent binding sites for GR and pSer2 Pol2 at putative gene targets with potential physiological significance. However, there is relatively little by way of new mechanistic insight, and further, no effort is made to analyze or discuss the current findings in the context of earlier published studies, including the author’s own prior work on CORT/GR pattern-dependent responses.

1) New mechanistic insights are generally lacking but could be obtained by using the current set of livers to elucidate differences in chromatin states between pulsatile vs. constant CORT exposure, including differences in chromatin accessibility, (c.f., Ref. #28) and/or relevant activating and repressive histone marks. The addition of such data is considered critical.

2) The major (new) findings here relate to differential gene targets of GR and pSer2 Pol2 between pulse and constant CORT. Two types of important supporting gene expression data are lacking: (A) Mechanistic Intronic RNA-seq analysis (or GRO-seq, etc.) is needed to determine if, in fact, there are significant differences in transcriptional rates between pulse and constant CORT for the differential pathways shown in Fig. 4ABC; and (B) Biological validation of result – RNA-seq of livers given Pulsatile vs. Constant CORT for a few days, to ascertain whether the differences in expression predicted by Fig. 4ABC are, infact, realized as ‘transcriptional outcomes’.

3) All raw high throughput data sets should be uploaded to GEO or another such public database, and accession numbers with reviewer access are needed. In addition supplemental Excel files with annotated, processed data listing all ChIP-seq peaks, peak intensities, differential sites, etc. should be provided.

4) Manuscript needs to be carefully edited to revise or clarify/explain apparent discrenpancies.

For examples:

a) Constant CORT levels, on lines 131 vs. line 146.

b) Light cycle times: line 138 vs. line 446.

c) Doses of CORT are unclear, and are listed in various inconsistent/inappropriate units for dosing: micromolar (line 141), rate in ml (line 143).

d) percentages referred to in text (lines 242-243) not explicity seen in the values shown in the associated Table in Fig. 3E; kb units in text (line 289) vs. log 2 bp shown in Fig. 3H.

5) How does CORT infusion, even in the controls (line 455) impact the basal level of CORT and the background of GR binding sites.

6) ChIP-seq data appear to be based on n=6 biological replicates which were pooled prior to sequencing (line 529). Is that true? If so, it seriously comprises the ability to know how reliable the differential peaks are.

7) Lines 223-227. Authors should consider the possibility that combinatorial patterns of the 6 motifs in Fig. S3B are differentially associated with the 3 types of peaks. This can readily be done by clustering each set of sites based on the panel of motifs with appropriate statistical analysis.

8) Fig. 1B. Comment how representative this is of the 6 profiles. Show the other 5 profiles in Supplemental.

9) Fig. 2A-2F – Recluster across the sets of sites to illuminate subsets of sites. Y-axis label is unreadable.

10) Results should be discussed (compare and contrast) in the context of other liver biological systems that intersect with GR, and where epigenetic and transcriptional outcomes depend on the pattern of hormone stimulation, such as Growth hormone-activated Stat5 signaling to metabolic genes.

Reviewer #3: The manuscript entitled “Corticosterone pattern-dependent glucocorticoid receptor binding and transcriptional regulation within the liver” builds on work done previously in the Lightman and Hager labs concerning the effect of glucocorticoid secretion rhythms on binding of the glucocorticoid receptor (GR) to DNA and consequent gene regulation. The existence of a pulsatile secretion rhythm of corticosterone (cort) on top of an overall diurnal rhythm was described previously in rats, and an effect of this pulsatile rhythm on GR binding and gene expression in cell culture has also been described. This work represents a step forward as the study was performed in a whole animal (rat) with the observed differences in a relevant tissue (liver). The administration and monitoring of cort in a pulsatile fashion, and in measuring differences in GR binding and gene regulation at corresponding intervals is a laudable technical achievement, and represents an important advance in understanding the role of glucocorticoid pulses in biology. The finding that certain unfavorable metabolic pathways are preferentially regulated by continuous administration of cort may also help to explain how dysregulation of glucocorticoids or pharmacological administration of glucocorticoids results in metabolic issues.

The collection of data and processing, in general, are solid, as is the overall interpretation. The experiments are also well reasoned. The authors do not make a strong conclusion that pulsatile exposure to glucocorticoids will lead to a more favorable metabolic profile than continuous administration (for example by continual regulation of cholesterol or carbohydrate metabolism), but this is appropriate, as the data are correlative, with no validation of the findings. This reviewer is strongly impressed by the system, and the demonstration that it shows something consistent with known biology. The lack of a strong conclusion, a comparison with results in cell culture systems, and follow-up to validate the links between differential binding/gene regulation and biology reduces the appeal to a broader audience.

Although the data collection and analysis appears solid, I have a few questions.

1) The measurement of cort in circulation every 10’ minutes is great. However, does the concertation in liver fluctuate by precisely the same period, or is it slightly out of phase? Measuring the concentration of cort in liver tissue at harvest timepoints could resolve this issue. It could also answer the question of whether the liver absorbs cort exactly the same way whether it is administered in a pulsatile or constant fashion.

2) Venn diagrams showing the overlap of genes or binding loci rely on arbitrary cutoffs, and can sometimes be misleading, or tell an incomplete picture. A scatterplot cmparing the normalized occupancy of each region under two different conditions would answer questions like: 1) Is GR bound to all of the same sites in PEAK_180 compared to PEAK_140, just less so? The answer to this would be very different than PEAK_180 GR being bound to only a subset of PEAK_140 sites. 2) Is GR bound to all the same sites in CONS_140 and CONS_180, but with just a few changes in the uncertainty of occupancy? This would be more informative of the biology of GR binding than saying that CONS_180 binds more sites (which is a distinct possibility). In other words, scatterplots can show trends in the data that are obscured by simple Venn diagrams that may be informative. For example, it is my hypothesis (which could be wrong) that the PEAK_140 binding pattern is simply more intense CONS_140 because there is, at that time point, more cort in the liver. If a scatterplot showed this (and quantified by simple linear regression), this could then be tested by increasing the CONS dose to match the instantaneous concentration in PEAK to see if there is a difference in binding pattern.

3) The authors do a beautiful job measuring GR occupancy and gene regulation (using Pol2 occupancy as proxy). The patterns of binding are actually fairly similar between PEAK and CONS, with PEAK having more peaks, but the expression patterns are wildly different – with pulsatile strongly biasing toward repression - yet the authors present no model as to how more or transient binding favors repression. Is the same behavior seen at early timepoints for GR in other systems? Does it suggest a kinetic model where the early effects of GR are largely repressive? This is a very interesting finding that is largely not discussed.

4) Related to the point above: Although there are caveats to such analyses, the authors make no attempt to correlate binding at specific loci with regulation of proximal or associated genes. This might help to explain the mechanism of why repression is favored by pulsatile administration whereas gene activation is favored by long-term administration.

I have a few general comments to improve the manuscript.

1) I’m confused by the layout of the manuscript. The figure legends appear to be embedded within the results section. I don’t know if this is how PLOS Genetics requests it, but I was not always sure when I was reading a result and when I was reading a figure legend. I’d prefer the figure legend separate from the results text.

2) The Figures, labels, and legends are not always clear:

a. Figure 1E – The ampersands, dollars and starts above the bars are not explained in the legend.

b. Figure 2 – the scale for 2A is “0 to 5”. Zero to five what? Fold enrichment? Normalized enrichment? This is not clear. Also, the y axis label at the far right is unreadable.

c. Figure 2: Often, and in other Lightman/Hager papers, it has proven useful to order each binding region heatmap by normalized tag intensity, or to cluster the tag intensities in order to reveal patterns of difference between samples. By eye it appears that there are clearly different clusters (e.g. about 2/3 of the way up comparing CONS-140 to CONS_180).

d. I couldn’t find a reference to 2J in the text, and it is a confusing comparison. Is this needed? Perhaps it could be replaced by Figure S3B, which is a more informative figure?

e. The sites are percentages at the bottom of Figure 3A-D are not explained in the legend, and are not intuitive to understand.

f. The lines in Figure 3G are unreadable – I can’t tell whether to accept the conclusion made in the text or not.

g. Similar to Figure 2J, Figure 3F is confusing, and is only used in the text to say that the biggest group of genes is the PULS_140.

h. Figure 3H is pointed out as an interesting finding, but no conclusion is drawn, so it’s hard to know why it is interesting or what it might imply.

i. Figure S1 has no scale bar, though the legend says it does.

j. Figure S2 is unreadably small, and when there are two samples in one plot it is not possible to tell which is which. If there are no limits on supplementary figure size, it would be advisable to make this much bigger with darker, easier to distinguish lines.

k. As noted above, Figure S3B is an important enough result to include in the main paper.

l. Any graph (Figure 4D,E,F, S4, S5) should have an x-axis label and scale.

3) The terminology “pattern dependent” in reference to how GR binds or which genes GR regulates, is sued throughout the manuscript, including the abstract, but it is not clear to me what it means. I do not think that this phrase clearly communicates that some binding and some gene regulation is dependent on pulsatile versus constant cort administration, if that is the meaning.

4) Line 80 of the intro – the authors claim that GC side effects can result from “even low doses for a prolonged period”. What is meant by low doses? Doses below the dose equivalent of what is secreted endogenously? How do low doses (e.g. 20mg) of prednisone compare to the endogenous levels of cortisol secreted? My back of the envelope calculation would indicate that this would represent a huge increase. The authors should be more explicit, as their manuscript relies on the idea that low dose administration would be at or below endogenous, pulsatile levels.

5) Line 116 of intro – though MetS is listed as an abbreviation at the end, it is not defined in the intro prior to its use.

6) Line 124 – what is “circadian active phase”?

7) Line169 – The authors say that CONS_140 has a “slightly reduced number” of sites compared to PEAK_140. It has 1273 sites compared to 2658, which is less than half. That is not slightly reduced.

8) For NADIR_180, the authors find not significant peaks over vehicle control, though by eye the tag density is clearly greater. NADIR_180 and vehicle are different, as is evidenced when NADIR_180 is used as a control and fewer peaks are identified in PEAK_140. This may be a case of a fold change cutoff obscuring identification of a statistically significant increase in binding. Indeed, there is still a substantial amount of CORT in circulation at the 180 time point – one would think that GR should still be bound to some loci.

9) Line 199 – it is not clear whether the motif analysis in Figure 2L/M was performed on a single sample or across all samples. This should be clarified in the legend.

10) Lines 202-206 – the authors state that there is unliganded GR binding to chromatin. To make the claim that there is no cort, the authors would have to test for cort concentration in the liver even and show that there is none even in adrenalectomized rats. Cort is synthesized in other organs, such as the thymus (see the work of John Ashwell), in mice. It may also be in rats.

11) Line 217/218 – CEBPA and C/EBP should be consistent.

12) Line 228-230. The Hager lab and others have never found evidence of the nGRE motif in GR ChIP-seq data sets. Only the Chambon lab has found this, with questionable in vitro support by the Ortlund lab. The notion that this nGRE is real has been debunked and should no longer be discussed as a motif that directs GR binding. Repeating it here gives it continued life.

13) The authors use Pol2 ChIP as a proxy for gene expression – this is great. Perhaps it would make the paper more readable, then, to refer to enrichment or depletion of Pol2 at *genes* rather than *sites* in order to be sure that reader stays cognizant that this is the conclusion you are trying to draw?

14) Line 315 – “patterned time points” – I don’t know what this means.

15) Line 360 paragraph – this paragraph is more interpretation than presentation of results, and should be moved to the discussion. Same with the paragraph after that. Also – line 372, it is just hard to follow the logic – are the authors saying that CONS administration of cort may impair VLDL transport due to persistent repression of ApoB? I think so, but it should be stated more definitively.

16) Line 384 – The author states that all of the binding present at PEAK_140 “dissociates completely” at NADIR_180. The data do not support this. A visual observation of NADIR_180 tag density versus vehicle (Figure 2) indicates that there is *some* binding at NADIR_180. In addition, there are >2,600 binding sites in PEAK_140 compared to vehicle, but only ~1500 in PEAK_140 when compared to NADIR_180. This discrepancy can most easily be explained by residual binding of GR at NADIR_180. This may be a case of fold change cutoff producing a misleading interpretation.

17) Line 402 – The author claims that CONS administration “activates” the cholesterol biosynthetic pathway. This is not supported by Figure 4. Figure 4 shows that pulsatile administration strongly represses the pathway, whereas CONS still represses, but more mildly. This is a “failure to fully repress”, not an activation of this pathway.

18) Line 429 – “chronobiological” is used to describe the effect of fluctuating GR levels on GR binding and gene regulation. This term is most frequently used to describe circadian or diurnal patterns. As the authors are attempting overlay a functional consequence to an ultradian rhythm on top of this circadian rhythm, I find this term confusing.

**Have all data underlying the figures and results presented in the manuscript been provided?**

Reviewer #1: **No: **NGS data has not yet been uploaded to a public database such as GEO

Reviewer #2: **No: **

Reviewer #3: **No: **The ChIP-seq data was not provided. Raw reads and/or at least peak tables should have been provided.

PLOS authors have the option to publish the peer review history of their article (what does this mean?). If published, this will include your full peer review and any attached files.

Reviewer #1: No

Reviewer #2: No

Reviewer #3: No

---

## [Decision Letter · Decision Letter 1]

21 May 2021

Dear Dr Flynn,

Thank you very much for submitting your Research Article entitled 'Corticosterone pattern-dependent glucocorticoid receptor binding and transcriptional regulation within the liver' to PLOS Genetics.

The manuscript was fully evaluated at the editorial level and by independent peer reviewers. The reviewers appreciated the improvements to the manuscript made upon revision. Reviewer #2 had some remaining concerns that we ask you address in a revised manuscript. I have already resolved reviewer #2 concerns about data submission and replicates with the reviewer, so please disregard those comments. Though, please update the manuscript text to reflect the submitted datasets (line 564), and consider further clarifying the description of the replicate structure around line 491. The remaining concerns to address from reviewer #2 are (2b, 2c, 2d) and (Prior comment #9). 

[LINK]

Yours sincerely,

Timothy E. Reddy

Guest Editor

PLOS Genetics

Wendy Bickmore

Section Editor: Epigenetics

PLOS Genetics

I have already resolved reviewer #2 concerns about data submission and replicates with the reviewer, so please disregard those comments. Though, please update the manuscript text to reflect the submitted datasets (line 564), and consider further clarifying the description of the replicate structure around line 491. The remaining concerns from the reviewer to address are (2b, 2c, 2d) and (Prior comment #9). If addressed, I expect that the manuscript would be suitable for publication.

Reviewer's Responses to Questions

**Comments to the Authors:**

Reviewer #1: The authors have addressed my comments.

Reviewer #2: Overall, the study is still lacking in mechanistic insight. The authors have been only partially responsive to prior comments.

At a minimum, the following three critical remaining issues should be addressed by textual revisions to the manuscript files:

Prior comment #2:

All raw high throughput data sets should be uploaded to GEO or another such public database, and accession numbers with reviewer access are needed. In addition supplemental Excel files with annotated, processed data listing all ChIP-seq peaks, peak intensities, differential sites, etc. should be provided.

2a) GEO files with a Reviewer accession link are still missing but required

2b) All Supplement tables (Excel files) require detailed table legends, so the reader can understand what information is presented in each column; in many cases this is cryptic. Transparency of underlying data is essential for any study of this sort.

2c) Supplementary tables should be numbered and cross-referenced in the Methods and/o results, as appropriate.

2d) Detailed figure legends are needed for all Supplemental figures

Prior comment #5:

ChIP-seq data appear to be based on n=6 biological replicates which were pooled prior to sequencing (line 529). Is that true? If so, it seriously comprises the ability to know how reliable the differential peaks are.

Author's revision to the text on lines 486-491 is still unclear. Were n=6 biological replicates (liver ChIps) analyzed, or were all six samples pooled prior to sequencing, resulting in an effective n=1 per condition/time point? The difference in terms of statistics is huge, but this important detail is still unclear. I guess that will become clear once the GEO accession link is available, but there is no reason the text should not disclose these details to the reader.

Prior comment #9

Results should be discussed (compare and contrast) in the context of other liver biological systems that intersect with GR, and where epigenetic and transcriptional outcomes depend on the pattern of hormone stimulation, such as Growth hormone-activated Stat5 signaling to metabolic genes.

Authors need to discuss their present findings in the context of other, recent studies of parallel biological systems of pulsatile hormone action in the liver. Authors' response basically declines to do so, despite the benefit of placing their work into a larger biological context. This is surprising, given the mechanistic insight other related studies may provide into the actions of GR, especially given the lack of mechanistic perspective here noted by all three reviewers

Reviewer #3: Thank you for the thoughtful responses to the comments. Though the responses did not always directly answer the concerns, the overall manuscript is much improved and I have no further comments.

**Have all data underlying the figures and results presented in the manuscript been provided?**

Reviewer #1: None

Reviewer #2: **No: **

Reviewer #3: Yes

PLOS authors have the option to publish the peer review history of their article (what does this mean?). If published, this will include your full peer review and any attached files.

Reviewer #1: No

Reviewer #2: No

Reviewer #3: No

---

## [Editor Report · Decision Letter 2]

23 Jul 2021

Dear Dr Flynn,

We are pleased to inform you that your manuscript entitled "Corticosterone pattern-dependent glucocorticoid receptor binding and transcriptional regulation within the liver" has been editorially accepted for publication in PLOS Genetics. Congratulations!

Yours sincerely,

Timothy E. Reddy

Guest Editor

PLOS Genetics

Wendy Bickmore

Section Editor: Epigenetics

PLOS Genetics

Comments from the reviewers (if applicable):

**Data Deposition**

http://datadryad.org/submit?journalID=pgenetics&manu=PGENETICS-D-20-00887R2

**Press Queries**

---

## [Editor Report · Acceptance letter]

5 Aug 2021

PGENETICS-D-20-00887R2 

Corticosterone pattern-dependent glucocorticoid receptor binding and transcriptional regulation within the liver 

Dear Dr Flynn, 

We are pleased to inform you that your manuscript entitled "Corticosterone pattern-dependent glucocorticoid receptor binding and transcriptional regulation within the liver" has been formally accepted for publication in PLOS Genetics! Your manuscript is now with our production department and you will be notified of the publication date in due course.

With kind regards,

Livia Horvath

PLOS Genetics

On behalf of:
